

# A probabilistic approach to quantifying soil property change
# through time integration of energy and mass input
Christopher Shepard[1][*], Marcel G Schaap[1], Jon D Pelletier[2], Craig Rasmussen[1]
[1]Department of Soil, Water and Environmental Science, The University of Arizona, Tucson, AZ,
USA, 85721-0038
[2]Department of Geosciences, The University of Arizona, Tucson, AZ, USA 85721-0077
*Correspondence to*: Christopher Shepard, 1177 E Fourth St, Room 429, Shantz Building, The
University of Arizona, Tucson, AZ, 85721-0038





**Abstract**

Soils form as the result of a complex suite of biogeochemical and physical processes;

however, effective modeling of soil property change and variability is still limited, and does not
yield widely applicable results. We suggest that predicting a distribution of probable values
based upon the soil-forming state factors is more effective and applicable than predicting discrete
values. Here we present a probabilistic approach for quantifying soil property variability through
integrating energy and mass inputs over time. We analyzed changes in the distributions of soil
texture and solum thickness as a function of increasing time and pedogenic energy (effective
energy and mass transfer, EEMT) using soil chronosequence data compiled from literature.
Bivariate normal probability distributions of soil properties were parameterized using the
chronosequence data; from the bivariate distributions, conditional univariate distributions based
on the age and flux of matter and energy into the soil were calculated, and probable ranges of
each soil property determined. We tested the ability of this approach to predict the soil properties
of the original soil chronosequence database, and soil properties in complex terrain at several
Critical Zone Observatories in the U.S. The presented probabilistic framework has the potential
to greatly inform our understanding of soil evolution over geologic time-scales. Considering
soils probabilistically captures soil variability across multiple scales and explicitly quantifies
uncertainty in soil property change with time.







## 1. Introduction


The need for pedogenic models that can be widely applied and easily utilized is
paramount for understanding soil-landscape evolution, soil property change with time, and
predicting future soil conditions. A mathematically simple, easily parameterized approach has
yet to be developed that is capable of predicting current soil properties or recreating potential soil
evolution with time. Here we address this knowledge gap through development of a probabilistic
model of soil property change capable of predicting soil properties across a wide range of
terrains, climates, and ecosystems.
The state factor approach has been one of the primary pedogenic models since it's
development in the late 1800's and early 1900's (Dokuchaev, 1883; Jenny, 1941). The soil state
factor approach (Jenny, 1941) assumes the state of the soil system or specific soil properties (S)
may be described as a function of the external environment, represented by climate (cl), biology
(o), relief (r), parent material (p), and time (t): S = f(cl, o, r, p, t). This approach increased our
understanding of soil variation across each factor, but more complex, multivariate approaches are
generally not possible or difficult to derive from this formulation (Yaalon, 1975). From the
original state factor model have evolved pedogenic models that include functional approaches
(Jenny, 1961), energetic approaches (Rasmussen and Tabor, 2007; Rasmussen et al., 2005, 2011;
Runge, 1973; Smeck et al., 1983; Volobuyev, 1964), and mechanistic approaches (Finke, 2012;
Minasny and McBratney, 1999; Salvador-Blanes et al., 2007; Vanwalleghem et al., 2013).
However, many of these approaches are either limited to a site-specific basis, require a high
degree of parameterization, or lack wide-scale applicability.
Here we develop a simple probabilistic approach to predict soil physical properties using
a large dataset of chronosequences studies. The model compresses state factor variability into 2



key components (parent material and total pedogenic energy, defined in Section 1.1) that were
parameterized and calibrated using the chronosequence database. Additionally, we modified the
model to include soil depth to capture the influence of redistributive hillslope processes to
predict soil properties. We hypothesized that by including soil depth, the model would
effectively predict the clay content in an independent dataset synthesizing soil and landscape
variability in complex, hilly terrain from a wide range of environments.

**1.1 Probabilistic model of soil property change**
The model presented here is based on a reformulated state-factor model, where a location
has a probability of displaying a range of differing soil morphologies and properties based upon
the state factors, with some range of values more probable than others, meaning the state-factor
model (Jenny, 1941) may be restated as:
$$\mathbb{P}(s_1 \leq S \leq s_2) = f(cl, o, r, p, t) \qquad (1)$$
where, the left hand side of the equation, $\mathbb{P}(s_1 \leq S \leq s_2)$, represents the probability that a given
soil will have a value located between a lower limit ($s_1$) and an upper limit ($s_2$) (Phillips, 1993b).
Eq. 1 can be restated more simply as:
$$\mathbb{P}(s_1 \leq S \leq s_2) = f(L_o, P_x, t) \qquad (2)$$
where, the original soil forming state factors have been simplified to represent the fluxes of
matter and energy into the soil system ($P_x$), incorporating the influence of climate and biology,
and the initial state of the soil forming conditions ($L_o$), incorporating the influence of the initial
topography and original soil parent material, and time or age of the soil system (t) (Jenny, 1961).
Equation 2 was further simplified to make the approach operational. A quantitative
measure of climate and biology was needed to represent the influence of $P_x$ on soil formation.





We used a quantification of $P_x$ calculated from effective precipitation and biological
productivity, termed effective energy and mass transfer (EEMT, J m$^{-2}$ yr$^{-1}$)(Rasmussen and
Tabor, 2007; Rasmussen et al., 2005, 2011). EEMT provides a measure of the energy transferred
to the subsurface, in the form of reduced carbon from primary productivity and heat transfer
from effective precipitation, which has the potential to perform pedogenic work, e.g., chemical
weathering and carbon cycling. Using EEMT as a simplification of $P_x$, Eq. 2 was restated as
(Rasmussen et al., 2011):
$$\mathbb{P}(s_1 \leq S \leq s_2) = f(L_o, \text{EEMT}, t) \qquad (3)$$
We further simplified Eq. 3 by combining the flux term EEMT and the age of the soil system (t).
EEMT multiplied by the age of the soil system, i.e. EEMT*t, provides an estimate of the total
energy transferred to the soil system over the course of its evolution, referred to here as "total
pedogenic energy" (TPE, J m$^{-2}$). The TPE provides an estimate of $P_x$ that incorporates soil age,
thus Eq. 3 may be restated as:
$$\mathbb{P}(s_1 \leq S \leq s_2) = f(L_o, \text{TPE}) \qquad (4)$$
where at a certain point in time the probability of a soil property existing between $s_1$ and $s_2$ is a
function of $L_o$ and TPE. Explicitly including time in Eq. 4 through TPE partially captures
variation in soil property change attributable to topography and parent material. Soil residence
time may be directly related to landscape position through topographic control on soil production
and sediment transport/deposition (Heimsath et al., 1997, 2002; Yoo et al., 2007). Additionally,
parent material modulates soil residence time through control on soil depth (Heckman and
Rasmussen, 2011; Rasmussen et al., 2005), soil production, and sediment transport rates (Andre
and Anderson, 1961; Portenga and Bierman, 2011). The initial conditions of the soil forming
system ($L_o$) are never fully known; however, representing the state of the soil system as a



probable distribution of values, implicitly accounting for soil age, and not constraining the initial
soil forming conditions, Eq. 4 can be stated simply as:
$$\mathbb{P}(s_1 \leq S \leq s_2) = f(TPE) \qquad (5)$$
where the probability state of the soil, $\mathbb{P}(s_1 \leq S \leq s_2)$, bounded by a lower and upper limit, is a
function of one quantifiable variable.
Quantitatively realizing Eq. 5 required the use of predetermined joint probability density
functions parameterized with TPE and a selected soil physical property. Bivariate normal density
functions were calculated to determine the probability of a soil property range given a TPE
value. The bivariate normal density distribution (Ugarte et al., 2008) was calculated as:
$$f(x,y) = \frac{1}{2\pi\sigma_x\sigma_y\sqrt{1-\rho^2}} \exp\left(-\frac{1}{2(1-\rho^2)}\left[\frac{(x-\mu_x)^2}{\sigma_x^2} + \frac{(y-\mu_y)^2}{\sigma_y^2} - \frac{2\rho(x-\mu_x)(y-\mu_y)}{\sigma_x\sigma_y}\right]\right) \qquad (6)$$
where, $\rho$ represents the correlation coefficient, $\mu_x$ is the mean of TPE, $\mu_y$ is the mean of the
selected soil physical property, $\sigma_x$ is the standard deviation of TPE, $\sigma_y$ is the standard deviation
of the selected soil physical property. Using the bivariate normal density functions, conditional
mean and variance values were calculated given a value of TPE; the conditional means and
variances parameterized conditional univariate normal distributions for the selected soil physical
properties. The conditional mean (Ugarte et al., 2008) was calculated as:
$$\mu_{Y|X=x} = \mu_y + \rho\frac{\sigma_y}{\sigma_x}(x - \mu_x) \qquad (7)$$
where, $\mu_{Y|X=x}$ is the conditional mean soil property value given a value for TPE. The conditional
variance (Ugarte et al., 2008) was calculated as:
$$\sigma_{Y|X=x}^2 = \sigma_y^2(1 - \rho^2) \qquad (8)$$
where, $\sigma_{Y|X=x}^2$ is the conditional variance of the soil property given a value of TPE.

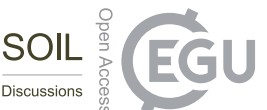

Applying this approach required certain assumptions and simplifications. The model
assumes that climate was constant over the entire duration of pedogenesis. The model makes no
assumptions about the progressive and regressive processes that drive pedogenesis; by weighing
all profiles equally, both progressive (e.g., horizonation, clay accumulation, reddening, etc.) and
regressive (e.g., haplodization, erosion, pedoturbation, etc.) pedogenic processes (Johnson and
Watson-Stegner, 1987; Phillips, 1993a), are implicitly captured in the model structure. The
model makes no assumptions about the initial soil forming system; the model simply describes
the probability of a location exhibiting a range of soil properties based on TPE. The model
assumes all changes in soil physical properties are due to pedogenic processes. We used a
bivariate normal distribution; consequently the model assumes the data conforms to a normal
distribution.

**2. Methods**
**2.1 Data collection and preparation**

The probability distributions were parameterized using an extensive literature review of

chronosequence studies. Over 140 chronosequence publications were identified using Google
Scholar (scholar.google.com) and ThomsonReuters Web of Science (webofknowledge.com),
forty-five of which contained the data required for inclusion within the present study. Inclusion
within the present study required: profile descriptions with horizon-level clay, sand, and silt
content, soil depth; well-defined ages of the soil-geomorphic surfaces; and geographic
coordinates or maps showing locations of the described profiles. The chronosequences spanned a
wide range of geographic locations, ecosystems, climates, rock types, and geomorphic landforms
(Fig 1, Table S1).




**2.2 Total Pedogenic Energy**

The influence of both climate and vegetation at the locations of each soil profile was
determined using effective energy and mass transfer (EEMT) (Rasmussen and Tabor, 2007;
Rasmussen et al., 2005). The EEMT values for each soil profile were extracted from a global
map of EEMT derived from the monthly global climate dataset of New et al. (1999) at 0.5°x0.5°
resolution using ArcMap 10.1 (ESRI, Redlands, CA) (Rasmussen et al., 2011). Total pedogenic
energy (TPE, J m$^{-2}$) was derived simply by multiplying EEMT (J m$^{-2}$ yr$^{-1}$) for each soil profile by
its reported age (yr). TPE was used because it was a better predictor of soil physical properties
relative to mean annual temperature, mean annual precipitation, or net primary productivity
(Table 3).

**2.3 Application to chronosequence data**

The chronosequence database included 45 distinct chronosequences representing 416
different soil profiles. We focused here on changes in sand, silt, and clay content and solum
thickness as proxies for soil change with time. We tested the approach on depth weighted (DWT)
sand, silt and clay content (reported as weight %), as well as the maximum measured value of
sand, silt, and clay content within each soil profile. Buried horizons were removed from the soil
profiles before either the maximum or DWT content values were calculated. For soils reported in
McFadden et al. (1986), surficial modern-aged eolian horizons were removed; the reported ages
of the soil-geomorphic surface more closely matched the buried horizons under the eolian
horizons. Solum thickness was extracted for each profile, defined as the thickness of the horizons
influenced by pedogenic processes or the depth to C horizons (Schaetzl and Anderson, 2005).



The site RW-14 from McFadden and Weldon (1987) was not included in the solum thickness
model calculations, the measured solum thickness of RW-14 was 1460 cm, an order of
magnitude greater than all other soil profiles included in the study. Four hundred and sixteen
profiles reported clay content data, only 398 profiles reported sand and silt content, and 410 soil
profiles contained a developed solum. We classified the soil profiles by parent material in terms
of igneous, metamorphic, or sedimentary and by geomorphic landform (e.g., alluvial surface,
marine terrace, or moraine, etc.) (Shoeneberger et al., 2012); for example, if a soil was formed on
an alluvial fan from granitic parent material, it would be defined as alluvial and igneous.

Using the soils data, we calculated bivariate normal probability distributions using TPE

and the soil physical properties (Eq. 6). The soil data were transformed using logarithmic and
square root transformations when appropriate to meet the normality assumption of the bivariate
normal probability distribution. Conditional univariate normal distributions (Eqs. 7, 8) were
calculated to approximate probable ranges of soil properties using leave one out cross validation
(LOOCV). Each of the soil chronosequences was removed from the model dataset, with the
remaining chronosequence data used to calculate the parameters of the bivariate and conditional
univariate normal distributions.  The conditional univariate normal distributions were calculated
using the TPE values for the profiles within the left-out chronosequence.

**2.4 Application to complex terrain**

By design, soil chronosequences are generally sited on gentle, low sloping terrain to

minimize the influence of topography and erosion/deposition on soil formation (Harden, 1982).
However, much of the Earth's surface is characterized by complex topography with high relief,
steep slopes, and differences in slope aspect. Any predictive soil model or approach must be



effective in both simple terrain and complex terrain. To test the ability of the model to predict
soil properties in complex terrain, we compiled data from upland catchments with variable parent
material and topography from the literature, as well as data available from the US NSF Critical
Zone Observatory Network (CZO, wwww.criticalzone.org) (Table 1) (Bacon et al., 2012;
Dethier et al., 2012; Foster et al., 2015; Holleran et al., 2015; Lybrand and Rasmussen, 2015;
Rasmussen, 2008; West et al., 2013). Data from several additional studies from complex terrain
were also included to test the model (Table 1) (Dixon et al., 2009; Yoo et al., 2007). These data
were accessed from: www.criticalzone.org, or Google Scholar (scholar.google.com). These
studies were included because they all contained horizon-level soil texture data, soil depth,
percent volume rock fragment data, and $^{10}$Be or U-series measures of soil erosion rates or
residence time, where mean residence time (MRT) was calculated as: MRT=h/E, where h is soil
depth (m) and E is erosion rate (m/yr) (Pelletier and Rasmussen, 2009b). We used published
coordinates to extract EEMT values, calculated from New et al. (1999), for each soil profile
using ArcGIS 10.1, and used EEMT and MRT to calculate TPE. It should be noted the coarse
resolution of New et al. (1999) EEMT values do not account for local scale variation in water
redistribution and primary productivity that can lead to significant topographic variation in
EEMT (Rasmussen et al., 2015). Using Eq. 6 and the parameters generated from the
chronosequence database, conditional mean depth weighted clay content was calculated for each
profile.
Due to the influence of redistributive hillslope processes on soil development (Yoo et al.,
2007), soil depth varies systematically across hillslopes (Heimsath et al., 1997); thus, soil depth
can be used to incorporate information about these processes within the model calculations. We



calculated the mass per area clay content of these profiles using soil depth to correct for these
processes, as:

$$\text{Mass per area clay (kg m}^{-2}) = (\rho_b)(h)\left(\frac{\mu_{Y|X=x,\text{DWT CLAY}}}{100}\right)\left(1 - \left(\frac{\text{RF\%}}{100}\right)\right) \qquad (9)$$

where, $\rho_b$ is the soil bulk density assumed to be 1500 kg m$^{-3}$ for all soil profiles, $\mu_{Y|X=x,\text{ DWT CLAY}}$
is the predicted conditional mean for depth weighted clay content (DWT CLAY) using Eq. 7,
RF% is the measured depth weighted percent volume rock fragments within the soil, when no
RF% data were available we assumed a value of 41.7%, which was the average RF% for profiles
with reported values, and h is the soil depth in meters. Using Eq. 9, mass per area clay was
calculated for each soil profile. Further, we examined the impact of depth, rock fragment
percentage, and predicted conditional mean DWT clay on the predicted mass per area clay
predictions using multiple linear regression.

**2.4.1 Coupling geomorphic model with probabilistic model**

Additionally, we applied the model independent of measured soil data, across a small

complex catchment in the Santa Catalina Mountains (Catalina-Jemez CZO, Fig 2a-b, Table 1)
(Holleran et al., 2015; Lybrand and Rasmussen, 2015). The ~6 ha catchment is located at an
elevation between 2300-2500 m with mixed conifer vegetation, approximately 30 km northeast
of Tucson, AZ (Fig 2, Table 1). The approach utilized soil depth and residence time output from
a process-based numerical soil depth model. The model used high resolution LiDAR derived
topographic data to estimate 2 m pixel resolution soil depth and erosion rates (Fig 2c) (Pelletier
and Rasmussen, 2009a). These data were coupled with topographically resolved EEMT values
that accounted for local hillslope scale variation in water redistribution and primary productivity
at a 10 m pixel resolution (Rasmussen et al., 2015) (Fig 2d). We used TPE based on modeled

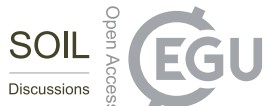

EEMT and soil residence time to predict DWT clay, and coupled with modeled depth in Eq. 9 to
predict mass per area clay at 2 m pixel resolution. We assumed a constant 50% rock fragment
value for each location. The coupled geomorphic-TPE model outputs were compared with point
measures of mass per area clay from Holleran et al. (2015) and Lybrand and Rasmussen (2015).
Model data were completely independent from Holleran et al. and Lybrand and Rasmussen and
these datasets served as a validation for the modeled output.

**3. Results**
**3.1 Application and parameterization to chronosequences**

The relationships between TPE and soil texture and solum thickness were used to

calculate the bivariate probability distributions. The bivariate probability distributions (Eq. 6)
were parameterized using the chronosequence database (Table 2). Furthermore, the relationship
between TPE and the soil properties was stronger than just using age, NPP, MAP, or MAT alone
(Table 3). Age was expected to strongly correlate to the soil properties due to the design of
chronosequence studies; however, comparing age and TPE separately, the percent increase in
Spearman rank correlations (r) ranged from 1.9% (DWT Silt) to 22.4% (Max Sand). Maximum
and depth weighted silt content were weakly correlated to both age and TPE and exhibited only a
minimal change in Spearman's rank correlation with TPE relative to age.

The correlation between TPE and maximum clay content (Fig 3, $\rho=0.78$, $r^2=0.61$,

$\sqrt{\text{Max Clay}} = -7.35 + 1.36 * \log(\text{TPE})$, df=414) was highly significant, and presented the
strongest probabilistic relationship determined between TPE and the soil properties. The
bivariate probability surface displayed the greatest probability around the joint means between
TPE and maximum clay content (Fig 3). Solum thickness and TPE were also strongly related,



but weaker relative to the maximum clay-TPE relationship (Fig S1, $\rho$=0.65, $r^2$=0.42,
$\log (\text{solum thickness}) = -0.57 + 0.27 * \log(\text{TPE})$, df=408). The relationships between TPE
and max sand (Fig S2) and silt (Fig S3) contents were generally weaker, relative to clay and
solum thickness, with little to no relationship between TPE and silt content.

285     The conditional univariate normal distribution parameters were determined for the soil

physical properties from the bivariate distribution and using Eqs. 7 and 8. The bivariate normal
distribution effectively predicted maximum clay content (Fig 4) with an $r^2 = 0.54$
(RMSE=14.7%) between the measured maximum clay content and predicted conditional mean
maximum clay content (Eq. 7) across all sites based on LOOCV (Fig 4d). The model effectively
predicted maximum clay content for each parent material with $r^2$ of 0.60 (RMSE=14.1%), 0.56
(RMSE=11.9%), and 0.59 (RMSE=16.7%), for igneous, metamorphic, and sedimentary parent
materials, respectively. The $r^2$ between the measured values and predicted values for solum
thickness, max sand, and max silt were 0.28 (RMSE=99.8 cm, Fig S4), 0.17 (RMSE=23.2%, Fig
S5), and 0.04 (RMSE=18.0%, Fig S6), respectively.

295     The relationship of predicted to actual maximum clay content varied significantly across

individual studies. The predicted values represent the predicted conditional means (Eq. 7)
bounded by the conditional standard deviation (Eq. 8), which approximates a 50% probability
that the measured maximum clay content will be within 1 standard deviation of the conditional
mean (Fig 5). The individual studies presented in Fig 5 were selected to represent a broad range
of climates and landforms, and demonstrate both the strengths and weaknesses of the model. For
Harden (1987) (Fig 5a, $r^2$=0.88, p<0.0001, df=20, RMSE=9.3%) and Howard et al. (1993) (Fig
5b, $r^2$=0.86, p<0.001, df=6, RMSE=9.8%), the model was generally successful at predicting the
maximum clay content values; both the Harden (1987) and Howard et al. (1993) sequences were



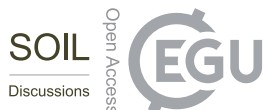

located in alluvial deposits but in vastly different climates, xeric (winter-dominated annual
rainfall regime) vs. udic (evenly distributed annual rainfall regime), respectively. The model was
capable of predicting maximum clay content values for glacial moraine deposits, in a frigid
climate (Fig 5c, $r^2$=0.87, p<0.0001, df=12, RMSE=6.1% Birkeland, 1984) and on marine terraces
in Northern California with a xeric climate (Fig 5f, $r^2$=0.98, p<0.001, df=4, RMSE=8.7%,
Merritts et al., 1991). The model was incapable of predicting clay accumulation on marine
terraces in hot, wet climates in Barbados (Fig 5d, $r^2$=0.31, p=0.08, df=9, RMSE=45.1% Muhs,
2001) or Taiwan (Fig 5e, $r^2$=0.67, p<0.001, df=11, RMSE=23.2%, Huang et al., 2010).

**3.2 Application in complex terrain**
The model was much less effective in complex terrain and highly overpredicted DWT
clay contents in soils located in complex landscapes (Fig 6a, $r^2$=0.26, y=0.39x+7.27, p<0.0001,
RMSE=5.3%). The model highly over predicted the clay content of the South Carolina site and
the Gordon Gulch soils, and under predicted the clay content of the Rincon, Santa Catalina,
Jemez sites.
When correcting for the influence of hillslope processes by explicitly including soil depth
and calculating mass per area clay, the approach effectively predicted clay content, with an
$r^2$=0.81 (Fig 6b, y=1.56x-15.2, p<0.0001, RMSE=84.4 kg clay m$^{-2}$), only slightly overpredicting
clay content, with a slope of 1.56. Soil depth was the strongest contributing factor to the mass per
area clay prediction with the greatest sums of squares in a simple multiple linear regression
including depth, RF%, and DWT clay% (Table 4); predicted conditional mean clay content
percentage was the second strongest contributing factor to the mass per area clay prediction.
Rock fragment percentage did not influence the mass per area clay content prediction.




### 3.3 Coupled geomorphic-TPE model

The coupled geomorphic-TPE model effectively predicted mass per area clay for the
majority of soils located within the Marshall Gulch subcatchment with an $r^2$=0.74 (Fig 7a,
y=0.85x-5.00, p<0.0001, RMSE=17.7 kg clay m$^{-2}$). For a subset of soils, the model did not
effectively predict mass per area clay, and were excluded from the regression in Fig 7a; four of
these soils were located on the east-facing ridge of the catchment, and an additional two soils
were formed on amphibolite rather than the granite or quartzite materials that all of the other
soils in the catchment were derived from. All of these locations also exhibited a poor fit between
modeled and measured soil depth (Fig 2e). The spatial distribution of mass per area clay was also
predicted across the catchment (Fig 7b), independently of measured data, and generally
conformed to previously predicted spatial distribution of clay stocks in the Marshall Gulch
catchment (Holleran et al., 2015).

### 4. Discussion

### 4.1 Model effectiveness

### 4.1.1 Model results for chronosequences

The model predicted maximum clay content across a diverse range of lithologies,
climates, and landforms. Weathering and clay production are primary pedogenic processes
(Birkeland, 1999; Schaetzl and Anderson, 2005), and because the model assumed all changes in
the soil profile are due to these processes, the model was the most effective at predicting clay
content. For initial soil states that begin pedogenesis with a potentially significant amount of
clay-sized particles the model was much less effective. The soils of the Taiwanese



chronosequence formed from conglomerates (Huang et al., 2010); conglomerates are typically
poorly sorted, such that these soils initially formed with high clay contents slowing clay
accumulation, limiting the effectiveness of the model to predict clay contents in these soils.
Additionally, the model highly underestimated the clay content of soils located on coral reef
terraces in tropical environments (Maejima et al., 2005; Muhs, 2001). Coral reef terraces
represent a relatively unique landform that weathers rapidly to fine sized particles, especially
under tropical climates, and generally have complicated parent material compositions (Muhs et
al., 1987). The combination of these factors limited the ability of the model to predict the soil
properties on these surfaces.
Sand and silt displayed weaker relationships with increasing total pedogenic energy. The
lack of correlation of sand and silt to TPE may result in part from the definitions of the particle
size classes. Sand sized particles span several orders of magnitude difference in particle size,
ranging from particles of 2 mm to 0.05 mm (Soil Survery Staff, 2010), whereas clays are
constrained to particles less than 0.002 mm. The sequential weathering of rock fragments and
coarse sand to fine and very fine sands therefore is not reflected in total sand content and likely
diminishes the relationship between sand content and total pedogenic energy and time (Pye and
Sperling, 1983; Pye, 1983; Sharmeen and Willgoose, 2006). The relationship between silt
content and pedogenic energy was the weakest of the three broad particles size classes (Tables 2,
3). Similar to sand, the silt size fractions span an order of magnitude in particle size ranging from
0.05 to 0.002 mm in diameter. Additionally, the silt fraction may also be heavily influenced by
deposition of eolian material and thereby introduce an additional mass of silt that was not
derived from the direct weathering of the initial soil forming system (McFadden et al., 1987)
effectively uncoupling silt content from total pedogenic energy.



Solum thickness displayed a relatively strong relationship with increasing pedogenic
energy, with TPE explaining up to 42% of the variance in solum thickness (Tables 2, 3). Soil
production is related to climatic variation (Amundson et al., 2015), with this variation partly
captured by EEMT and TPE, leading to the slightly stronger predictive power of the model.
However, soil production is also highly influenced by redistributive hillslope process, chemical
and physical weathering, and tectonic uplift (Heimsath et al., 1997; Riebe et al., 2004; Yoo and
Mudd, 2008b), and can be a highly non-linear process (Pelletier and Rasmussen, 2009a). These
factors were not directly accounted for in this study in that topography was not a quantified
factor, which likely represents a large proportion of the remaining unexplained variance in solum
thickness.

**384    4.1.2 Model results in complex terrain**

Due to using soil chronosequence data to parameterize the approach, the influence of
redistributive hillslope processes was not captured. Additionally, in the amount of time required
to transport soil across a hillslope, chemical and physical alterations of the soil particles are
possible and may not be reflected in mean residence time calculations (Yoo and Mudd, 2008a;
Yoo et al., 2007). Soil thickness is highly dependent upon hillslope position and landscape
morphology (Dietrich et al., 2003; Heimsath et al., 1997; Pelletier and Rasmussen, 2009a). By
using soil thickness as a proxy for the strength of these redistributive hillslope processes, and
converting the predicted conditional mean clay content value to a mass per area basis, the model
was able to capture differences in clay content across complex terrain for a variety of lithologies
and climates. The differing lithologies, climates, or vegetation types did not appear to impact the
ability of the model to predict clay contents, likely because soil depth accounts for many of these



controls. Parent material and climate influence the weathering process and production of clay in
soils (Harden and Taylor, 1983; Muhs et al., 2001); however, these factors are collinear with soil
depth (Heckman and Rasmussen, 2011; Lybrand and Rasmussen, 2015; Pelletier and Rasmussen,
2009a), such that by including soil depth, differences due to lithology or climate were partly
incorporated in the model prediction.

**4.1.3 Results from coupled geomorphic-TPE model**
For the majority of sites in the Marshall Gulch sub-catchment, the coupled geomorphic-
TPE model was highly effective at predicting clay content, and the spatial distribution of clay
stocks. Large differences were found for four soils located on the east-facing ridge of the
catchment underlain by granite with the model generally over-predicting soil depth and clay
content. Discrepancies between the modeled and measured depths were likely the primary
sources of error within the mass per area clay predictions for the 4 east-facing ridge soils (Fig
2e). The geomorphic model predicted deeper soil depths due to the presence of an apparent
convergent zone on the east-facing ridge of the sub-catchment; however, this convergent zone is
only a small feeder tributary to the larger catchment drainage. The inability of the model to
effectively predict clay contents and the mismatch between modeled and actual soil depths in the
4 soils located on the east-facing ridge is likely due to this local, fine-scale topographic variation
Error in predicted soil depths due to fine-scale differences in lithology within the
Marshall Gulch sub-catchment partly explains the discrepancies between measured and predicted
mass per area clay contents. For two amphibolite-derived soils, the model greatly underestimated
mass per area clay. The geomorphic soil depth model assumed a uniform weathering rate based
on the granitic soils (Pelletier and Rasmussen, 2009a); due to differences in primary mineral



assemblage, the amphibolite materials are likely weather at a faster rate compared to the granite
derived soils (White et al., 2001; Wilson, 2004), resulting in greater clay production and likely
explaining the underestimated clay contents. Inclusion of differential weathering rates for
varying lithologies within the geomorphic model would likely lead to better prediction of clay
contents. With these adjustments, the coupled geomorphic-TPE model represents an effective,
independent prediction of clay stocks.

**4.2 Advantages of probabilistic approach**

Simplifying and representing the soil-forming factors as multivariate distributions and

probabilities has the potential to quantitatively represent the general state-factor model, making
the approach universally applicable. The initial state of the soil can likely never be fully known,
leading to variability in soil properties over time that cannot necessarily, or ever, be attributed to
any external factor (Phillips, 1989, 1993b). A probabilistic approach utilizes that variability to
drive predictions and understanding of these systems. Similar to the approach taken here,
building distributions of the soil-forming state factors that are associated with distributions of
particular soil properties could yield probabilistic predictions of soil formation and change. We
selected to use a representation of climate and biology (EEMT), however, depending on the soil
property of interest the variables needed to parameterize the distributions would likely change;
for example, if interested in organic matter content, aboveground net primary productivity or
normalized difference vegetation index may be better predictors of organic matter accumulation.
The strength of this approach lies in the fact that no assumptions are made about the initial
conditions of the soil forming system or the specific soil forming processes. Predicting probable
distributions of soil physical properties implicitly acknowledges that our understanding of any



system is incomplete, but explicitly quantifies uncertainty in predictions and constrains the
potential observable values to a predicted range. Utilizing this approach will require the
necessary data to build distributions that are widely representative and applicable to most
locations (Yaalon, 1975). With wide accessibility to large databases of soil information, such as
the US National Soil Information System (NASIS) and the FAO Harmonized World Soil
Database, access to the required amount and quality of data may be possible. Similar to the
present study, simple bivariate distributions could be solved to calculate conditional distributions
based on the soil-forming state factors, effectively producing quantitative probabilistic
representations of Jenny's original equation (Jenny, 1941).

The simplicity of the present approach allows easy integration into pre-existing

geomorphic models of landscape evolution. Past approaches that have combined pedogenic and
landscape evolution models have generally focused on producing hypothetical soil-landscape
relationships that progress forward through time (Minasny and McBratney, 2001; Vanwalleghem
et al., 2013), or have focused on idealized landscapes (Temme and Vanwalleghem, 2015).
However, by combining probabilistic approaches parameterized using known landscapes, and
geomorphically based landscape evolution models, both potential soil-landscape evolution
scenarios can be investigated, as well as predictions of the current state of the soil-landscape. As
was demonstrated in Fig 7B, combining the present approach with geomorphically based soil
depth models generated from DEMs has great potential to predict soil properties across a diverse
range of environments, without needing prior knowledge of the landscape other than topography
and climate.

**4.3 Limitations and potential refinements**



There are obvious limitations within the current model: lack of consideration of parent
material influences, topographic variation, or internal soil feedbacks and thresholds, and
differences in paleoclimate variation. Parent material control on the relative proportion of
weatherable minerals and mineral weathering rates (Jackson et al., 1948) can manifest as vastly
different soil morphologies and rates of pedogenesis when controlling for other soil forming
factors (Heckman and Rasmussen, 2011; Parsons and Herriman, 1975). The current approach
implicitly assumes no information about the initial conditions, only that all clay production is a
pedogenic process. Applying this approach to parent materials, where a large fraction of clay-
sized particles formed through non-pedogenic processes, is thus limited and may explain why the
model was ineffective for some soils. Refining the current approach would require normalization
of soil to the particle size distribution of the soil parent material. Past studies have utilized highly
characterized parent material data to model soil property change with time (Chadwick et al.,
1990; Harden, 1982), but these data are generally difficult to obtain and often not reported in the
available chronosequence literature.
Topography dictates soil chemical and physical properties and residence times, especially
in complex terrain (Almond et al., 2007; Egli et al., 2008; Lybrand and Rasmussen, 2015), where
non-linear diffusive hillslope processes control the fluxes of matter and energy into and out of
the soil system (Heimsath et al., 1997; Pelletier and Rasmussen, 2009a; Rasmussen et al., 2015;
Yoo and Mudd, 2008b; Yoo et al., 2007). Using earlier versions of EEMT (Rasmussen and
Tabor, 2007; Rasmussen et al., 2005), the current formulation of the model and TPE does not
explicitly quantify topographic variation, which may account for error within current soil
property distributions and predictions. With the inclusion of topographic variation within EEMT
(Rasmussen et al., 2015) and topographic control of soil residence times (Foster et al., 2015;



West et al., 2013), we were able to correct this error within the present approach, particularly in
complex terrain, and effectively predicted clay stocks.

Internal or intrinsic feedbacks and thresholds within the soil system drive pedogenic

development without changes in the external state factors (Chadwick and Chorover, 2001; Muhs,
1984). For example, greater chemical weathering and clay production due to increased water
residence time caused by argillic horizon development is the result of an internal feedback that is
independent of the external climatic and biological system (Schaetzl and Anderson, 2005). These
thresholds can operate as progressive or regressive processes, driving soil formation forward or
hindering further development (Johnson and Watson-Stegner, 1987; Phillips, 1993a). Internal
soil development feedbacks were not explicitly considered in the present model formulation. The
presence of these internal feedbacks may partially explain error within the model predictions.
Changes in EEMT would not explain all observed differences in soil properties over the age of
the soil. However, if these feedbacks were operating in the included soils, the influence of
intrinsic thresholds was implicitly captured within the probability distributions, partially
accounting for the role of internal soil development feedbacks on soil formation.

Furthermore, global climate patterns have shifted dramatically over the last 65 Mya

(Zachos et al., 2001). The majority of soils observed in the compiled chronosequence database
span the Quaternary, including both the Holocene and Pleistocene. The Pleistocene was marked
by a number of major glacial-interglacial cycles at approximately 100,000-year intervals (Imbrie
et al., 1992; Wallace and Hobbs, 2006), which corresponded with shifting climatic conditions,
e.g., for large portions of the northern mid-latitudes glacial periods were generally cooler and
wetter, and interglacial periods were warmer and drier (Connin et al., 1998; Petit et al., 1999).
Further, the Pleistocene climate shifts likely influenced the rates of weathering and clay



production (Hotchkiss et al., 2000). Taking into account the differences in past and modern
climate would likely diminish disparities between observed and modeled soil physical properties.
Reconstructed global paleo-EEMT values would improve model accuracy, and limit uncertainty
in the probabilistic ranges of soil properties for soils older than Holocene age.

**5. Conclusion**

The present approach effectively predicts soil physical properties across a diverse range

of geomorphic surfaces, lithologies, ecosystems, and climates. Further, this approach is
mathematically simple and only requires knowledge of the probable age of a geomorphic surface
and the effective energy and mass transfer value associated with a given location, making this
approach universally applicable. The simplicity of the probabilistic approach is the lack of
assumptions about the initial conditions of the soil forming state or the processes driving soil
property change. A probabilistic approach does not exactly predict a soil physical property value
at a given location, but constrains the probable values based upon the state of the external
environment to the soil. Using probabilistic approaches, we can model probable soil-landscape
evolution scenarios, greatly informing our understanding of the evolution of critical zone
structure.

**Acknowledgements**
We thank Molly Holleran, Rebecca Lybrand, and Ashlee Dere for providing data for this study.
Support for C.S. was provided by the University Fellows program at the University of Arizona.
This research was funded by the U.S. National Science Foundation grant no. EAR-1331408





provided in support of the Catalina-Jemez Critical Zone Observatory. LiDAR data acquisition
was supported by U.S. National Science Foundation grant no. EAR-0922307 (P.I. Qinghua Guo).

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




**List of Figures and Tables**

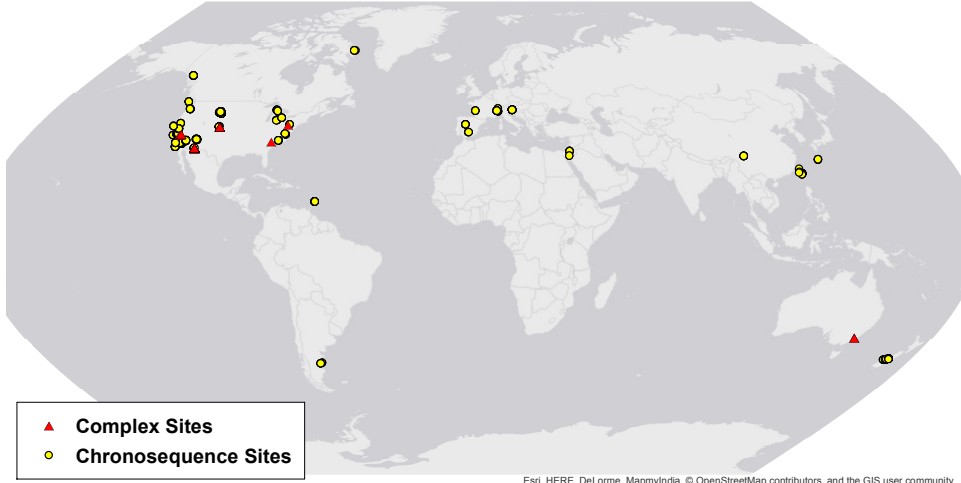

**Figure 1. Map of study sites.** Yellow points indicate location of chronosequences, and red triangles indicate location of soils in complex terrain.



**Table 1.** Complex terrain study sites and characteristics.

Table 1. Complex terrain study sites and characteristics

| Site | Study | Number of Sites | Elevation (m) | MAP (cm) | MAT (°C) | Parent Material | Slope | Aspect | Vegetation |
|---|---|---|---|---|---|---|---|---|---|
| Marshall Gulch Granite Subcatchment, Arizona, USA | Holleran et al., 2015. *SOIL.* 1:47-64. Lybrand and Rasmussen. 2015. *SSSAJ.* 79, 1: 104-116 | 24 | 2300-2500 | 85-90 | 10 | Granite, Amphibolite, Quartzite | 45% | North | *Pinus ponderosa, Pseudotsuga menziesii, Abies concolor* |
| Frog's Hollow, New South Wales, Australia | Yoo et al., 2007. *JGR.* 112: F02013 | 2 | 930 | 55-75 | ~16 | Granodiorite | - | - | *Ecalyptus* grassland savannah |
| Cross Keys, South Carolina, USA | Bacon et al., 2012. *Geology.* 40, 9: 847-850 | 1 | - | 115-140 | 14-18 | Granitic gneiss | <2% | - | *Quercus, Carya* |
| Gordon Gulch, Colorado, USA | Foster et al., 2015. *GSA Bulletin.* 127, 5/6: 862-878; Dethier et al., 2012. *Geomorph.* 173-174: 17-29 | 9 | 2440-2740 | 52 | 5 | Gneiss, Quartz monzonite, granodiorite | 15° - 28° | North and South | *Pinus ponderosa, Pinus contorta* |
| Rincon Mountains, Arizona, USA | Rasmussen, 2008. *Geochem. Cosmochem. Acta.* 72: A778. | 11 | 1050-2500 | <40-80 | 10-18 | Granodiorite(?) | - | - | Oak grass woodland, Piñon-Juniper woodland, Mixed Conifer |
| Jemez Mountains, New Mexico, USA | Huckle et al., 2016. *Chem. Geol. in press.* | 4 | 2990-3100 | ~50 | 4 | Rhyolite, tuff | - | West and East | *Pseudotsuga menziesii, Abies concolor, Picea pungens, Populus tremuloides* |
| Shale Hills, Pennsylvania, USA | West et al., 2013. *JGR: Earth Surf.* 118: 1877-1896; Ma et al., unpublished | 6 | 260-280 | 100 | - | Shale, sandstone | 15° - 20° | North and South | - |
| Sierra Nevada Mountains, California, USA | Dixon et al., 2009. *Earth Surf. Proc. Landf.* 34: 1507-1521 | 5 | 216-2991 | 37-106 | 3.9-16.6 | Tonalite, granodiorite | - | - | Oak-grass woodland, Mixed Conifer, Subalpine |



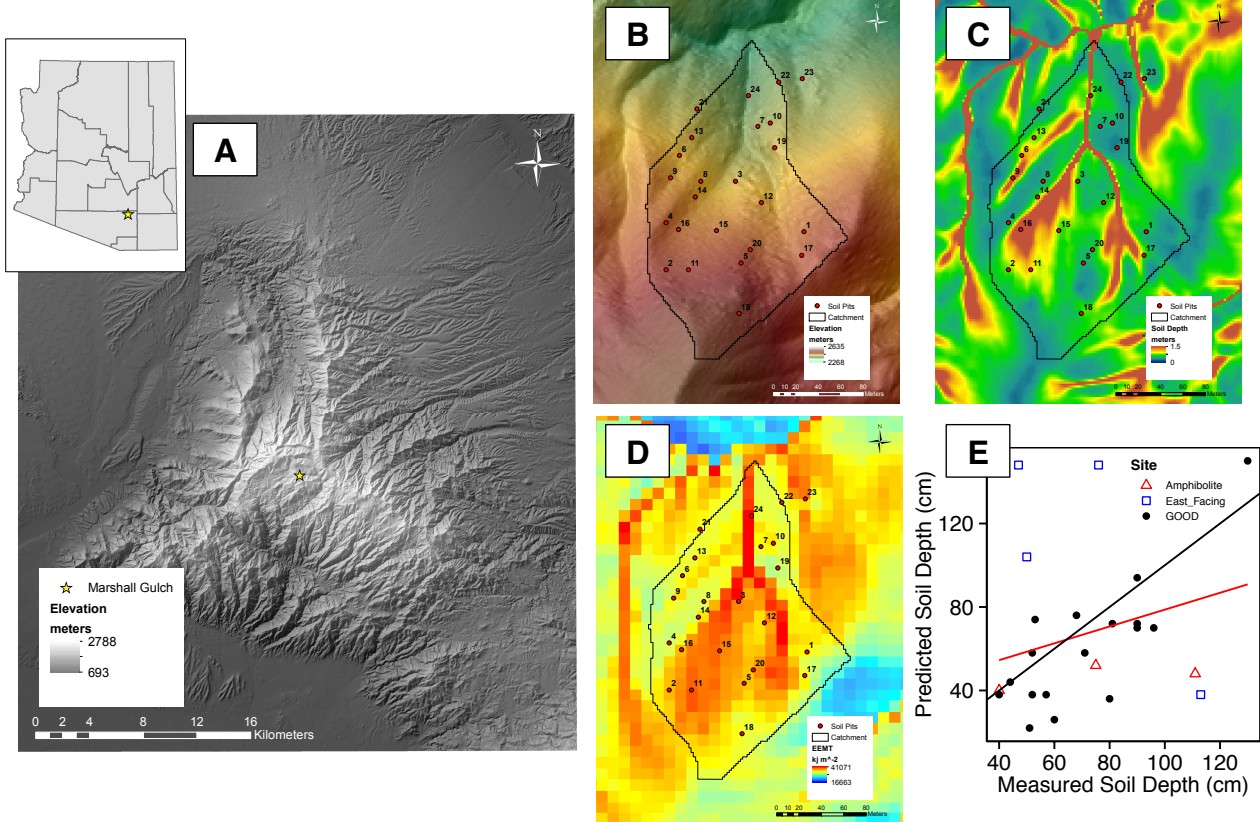

**Figure 2. Marshall Gulch study site.** (A) Location of the Santa Catalina Mountains and the Marshall Gulch catchment within Arizona, USA; (B) Elevation of the granite sub-catchment of Marshall Gulch; (C) Predicted soil depth in the granite sub-catchment (Pelletier and Rasmussen, 2009a); (D) EEMTv2.0 in the granite sub-catchment (Rasmussen et al., 2015); (E) Mismatch between the measured soil depths and predicted soil depths.




**Table 2.** Parameters for the bivariate normal probability distributions for the soil physical properties and TPE.

### Table 2. Soil property parameters

| Variable | n | μ | σ | ρ[a] |
|---|---|---|---|---|
| Max Sand | 398 | 70.51 | 25.39 | -0.48 |
| Max Silt | 398 | 34.80 | 18.38 | 0.32 |
| Max Clay[b] | 416 | 4.52 | 2.24 | 0.78 |
| DWT Sand | 398 | 59.03 | 26.04 | -0.57 |
| DWT Silt[b] | 398 | 4.55 | 1.68 | 0.26 |
| DWT Clay[b] | 416 | 3.66 | 2.09 | 0.73 |
| Solum Thickness[c] | 410 | 1.77 | 0.52 | 0.65 |
| | 416[d] | 8.71 | 1.29 | - |
| TPE[c] | 398[e] | 8.72 | 1.28 | - |
| | 410[f] | 8.73 | 1.26 | - |

[a]ρ, Pearson correlation between soil variables and Total Pedogenic Energy

[b]Square root transformed

[c]Log10 transformed

[d]For clay variables

[e]For sand and silt variables

[f]For solum thickness

n = number of profiles, μ = mean, σ = standard deviation

Max indicates maximum content

DWT indicates depth weighted average content



**Table 3.** Spearman rank correlations between soil physical properties and TPE and age.

| Variable | NPP | MAP | MAT | TPE | Age | % Increase[a] | n |
|---|---|---|---|---|---|---|---|
| Max Sand | -0.30 | -0.10 | -0.26 | -0.46 | -0.37 | 22.4 | 398 |
| Max Silt | -0.03 | -0.16 | 0.09 | 0.31 | 0.32 | -4.5 | 398 |
| Max Clay | 0.15 | -0.01 | 0.36 | 0.80 | 0.73 | 9.3 | 416 |
| DWT Sand | -0.22 | -0.04 | -0.29 | -0.57 | -0.50 | 14.1 | 398 |
| DWT Silt | 0.05 | -0.06 | 0.06 | 0.24 | 0.23 | 1.9 | 398 |
| DWT Clay | 0.21 | 0.02 | 0.39 | 0.75 | 0.67 | 12.4 | 416 |
| Solum Thickness | 0.12 | 0.06 | 0.22 | 0.63 | 0.57 | 10.4 | 410 |

Max indicates maximum content

DWT indicates depth weighted average content

[a]Precent increase in Spearman rank correlation between TPE and age




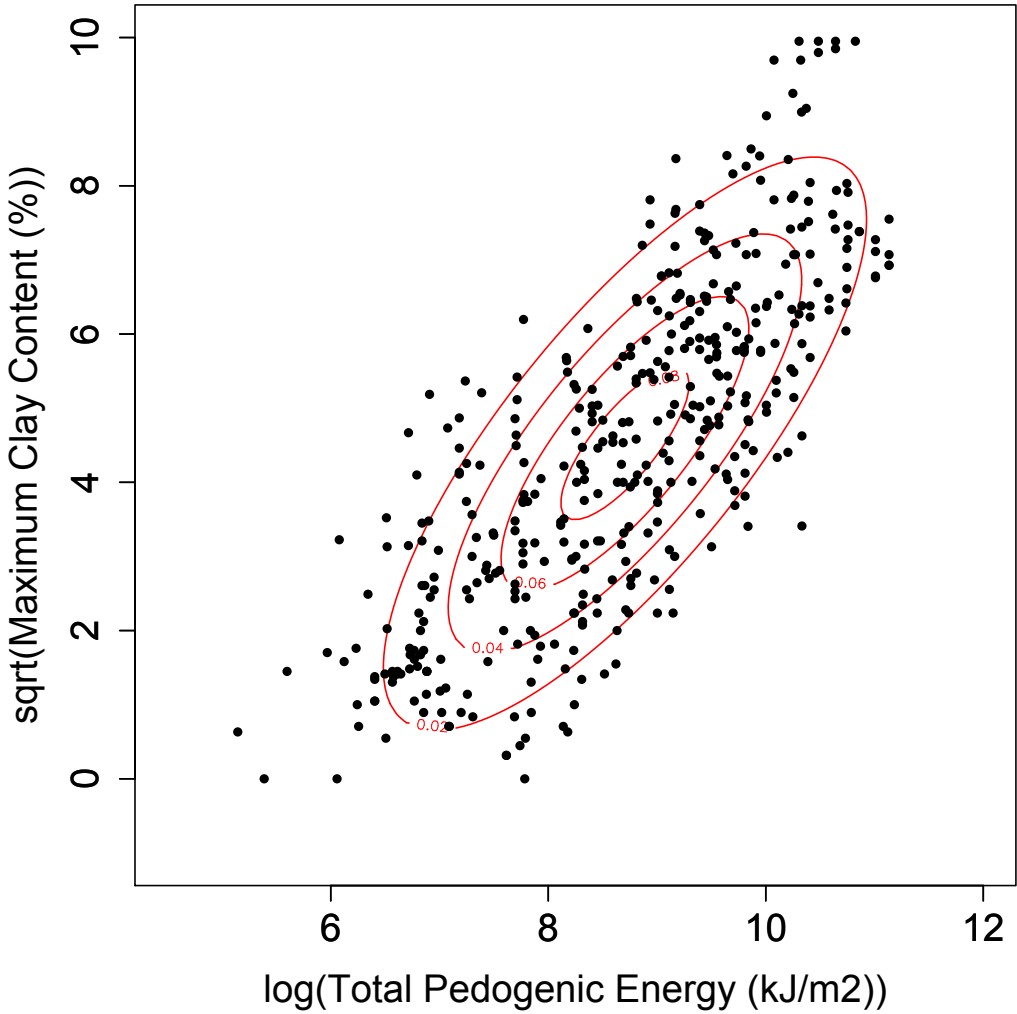

**Figure 3. Bivariate normal distribution between TPE and max clay content.** The points indicate individual soils. The red ellipses represent lines of equal probability, which corresponds to a three dimensional probability distribution. From this relationship the conditional mean and variances for the soil physical properties were calculated.




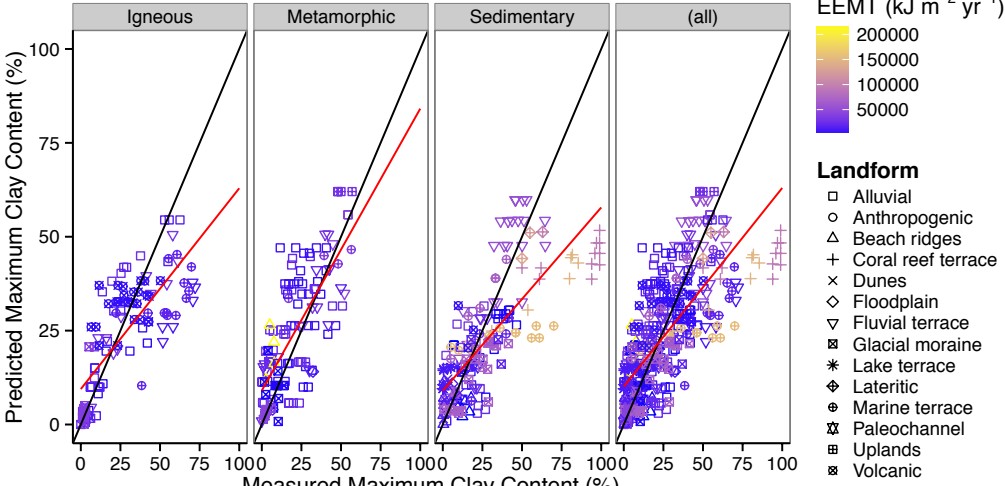

**Figure 4. LOOCV results for max clay content.** The results were subdivided by general soil parent material: igneous, metamorphic, and sedimentary; the points represent the geomorphic surface each soil formed on, and the colors represents the EEMT value for the location of each soil. Using LOOCV, where one chronosequence was removed from the model dataset and the remaining datasets were used to predict the parameters of the bivariate distributions, the conditional means of the left out chronosequence was determined. The model was effectively able to predict the conditional mean values of the max clay contents with an $r^2$=0.54 (RMSE=14.7%). The model was least capable of predicting the clay contents on coral reef terraces (+), and appeared the most effective for alluvial surfaces (□).




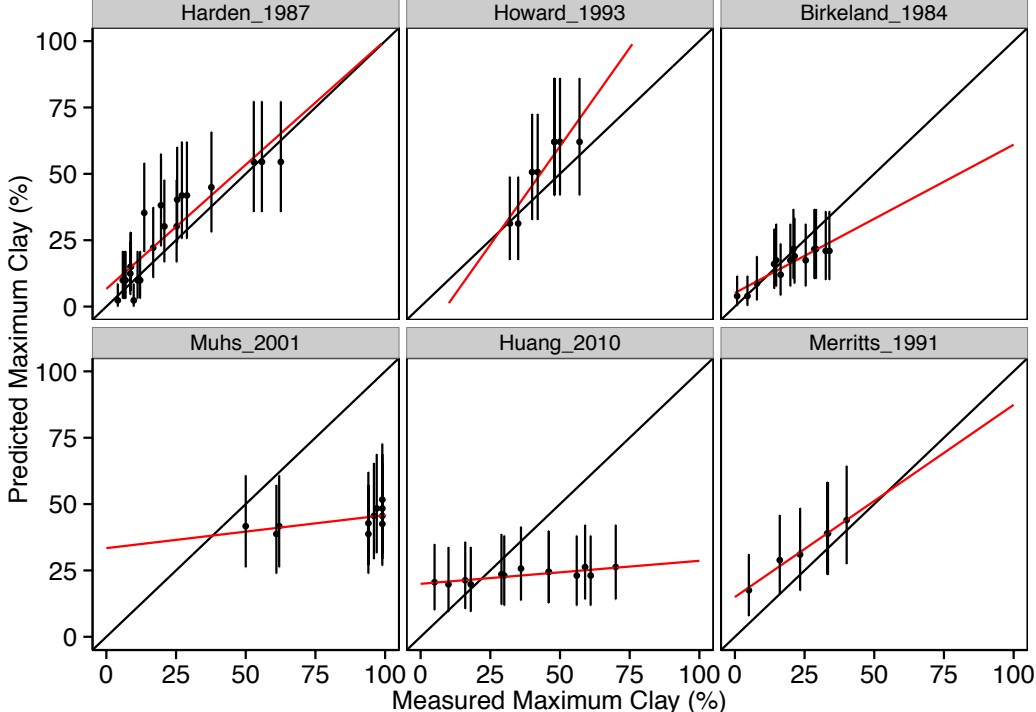

**Figure 5.** Selected relationships between the measured maximum clay content and predicted maximum clay content. A) Harden, 1987, B) Howard et al., 1993, C) Birkeland 1984, D) Muhs, 2001, E) Huang et al., 2010, and F) Merritts et al., 1991. The errors represent the conditional standard deviations around the mean, which correspond to a probability of 50%. The model effectively predicted clay content across a diverse range of climates, landforms, and parent materials. The model was the least effective at predicting the clay content of soils in tropical climates, and soils forming on coral reef terraces.




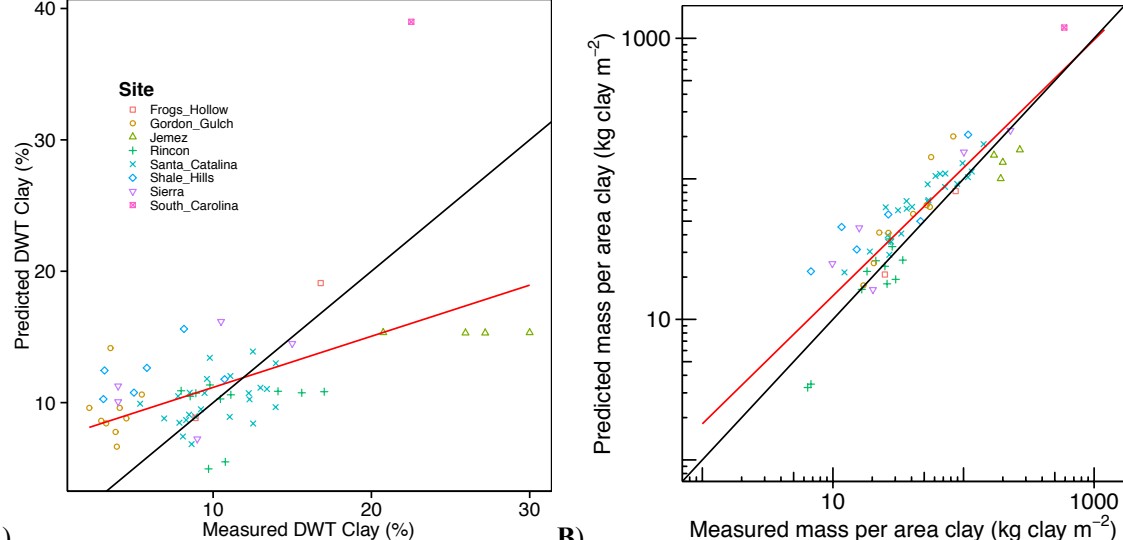

A)  B)

**Figure 6. Model results in complex terrain.** (A) Prediction of depth weighted (DWT) clay contents; (B) Prediction of mass per area clay using Eq. 9. The model was incapable of directly predicting DWT clay for the soils in complex terrain due to redistributive hillslope processes, $r^2$=0.25 between measured and predicted conditional mean DWT clay (A). By including information about soil depth and percent volume rock fragment, and converting DWT clay to mass per area clay, the model was significantly more effective at predicting clay contents for these soils $r^2$=0.81.



**Table 4.** Sensitivity analysis of model prediction in complex terrain.

**Table 4. Sensitivity analysis of model prediction in complex terrain**

| Effects | DF | Sums of Squares | Mean Sums of Squares | F value | p |
|---|---|---|---|---|---|
| Depth, $h$ (cm) | 1 | 1129156 | 1129156 | 469.0 | $< 2e\text{-}16$ |
| CM DWT Clay, $\mu_{Y|X=x}$ (%) | 1 | 142430 | 142430 | 59.2 | 2.0E-10 |
| Rock fragment, RF% (%) | 1 | 3013 | 3013 | 1.3 | 0.27 |
| Residuals | 58 | 139632 | 2407 | | |




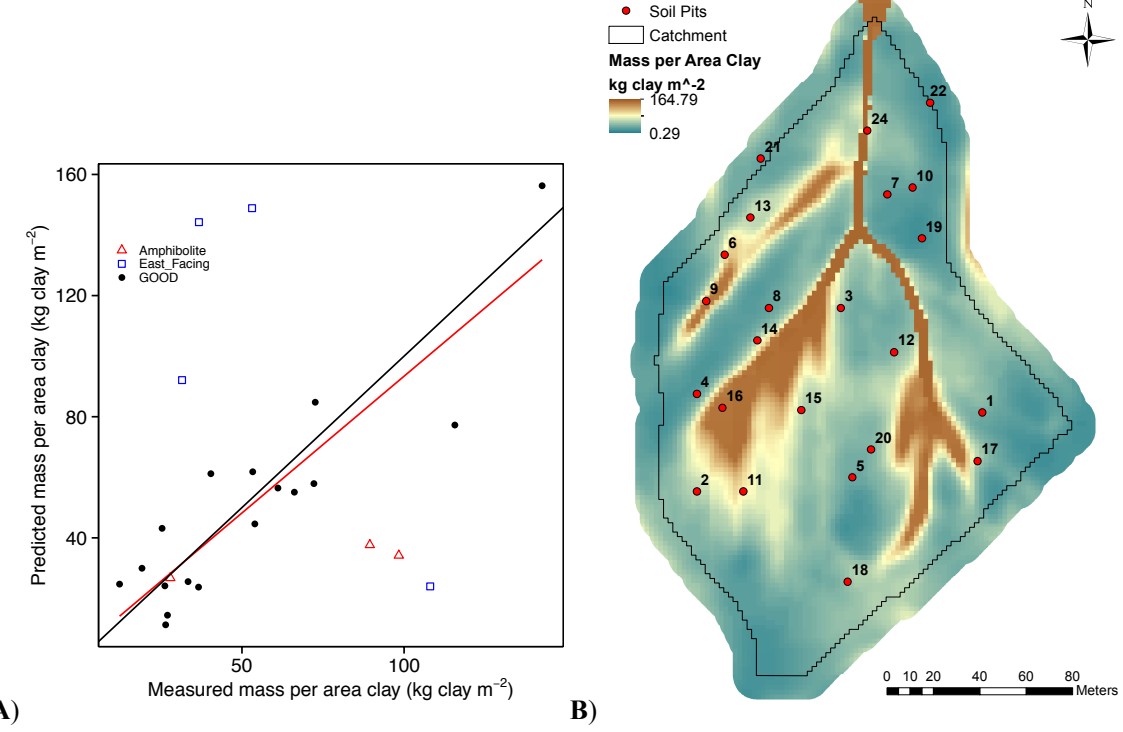

**Figure 7. Model results of coupled geomorphic-EEMT-TPE model in Marshall Gulch granite sub-catchment.** (A) Prediction of mass per area clay for sites from Holleran et al. (2015) and Lybrand and Rasmussen et al. (2015); (B) Spatial prediction of mass per area clay When combining the present approach, with a geomorphic based soil depth model, the combined models together were highly effective at predicting the clay contents for a majority of soils in the Santa Catalina Mountains (Catalina-Jemez CZO), $r^2$=0.74.