# Peer review of "A probabilistic approach to quantifying soil property change"

_SOIL, 2016_

## Editor Comment (EC1) · P.A. Finke (Editor) · 2 Nov 2016

Manuscript SOIL2016-63 "A probabilistic approach to quantifying soil property change through time integration of energy and mass input" by Shepard et al.

The authors present a probabilistic approach to estimate soil properties from energy and mass inputs accumulated over time into a parent material. As such this predictive model builds on the energy models from Runge and later instances by Rasmussen et al. It is parametrized and applied on a number of chronosequences and then onto CZN sites. As such model is not mechanistic in itself although it uses inputs that suggest a mechanistic nature (energy input), I have doubts on the use potential of such model in the CZN environment where process knowledge is being generated and applied.

At best, the model predicts some soil properties at unvisited locations where we have some idea on mass/energy fluxes and the age of the soil. In this respect it may have some value in the so-called homosoil approach in digital soil mapping, when practically no soil data are available. The model does however not generate knowledge on soil (forming) processes although the used wording in the manuscript suggests so; and there are (too) many assumptions behind it.

My conclusion is that this article needs a major revision in the sense that the below remarks need to be addressed and also, that the model application domain should be delineated more precisely (delineated in spatial and temporal extent and parent materials). I advise against keeping in the text that soil forming processes are addressed, the word "implicitly" actually means that only the net effect is dealt with, not any individual process unless perhaps weathering via EEMT and t.

Below I first formulate a number of general questions regarding the manuscript and then address some ambiguities or issues at specific locations in the manuscript. 1. Why is the model forced into a bivariate pdf form? Techniques were on the shelf?

2. Inputs "time" and "EEMT" are assumed to be known. EEMT can be assessed using today's data which brings in the (big) assumption that EEMT is a constant (section 1.1). All climate modelers will tell you that, e.g. due to variations in insolation, EEMT is not a constant. Glaciations were caused by this... Time or soil age is used for the integration. How often does one know the soil age, especially in non-glaciated areas? If it has to be measured on-site via dating techniques, this investment is far higher than direct measurement of the target variables, so why do it?

3. I am uncomfortable with the statements on removal of some soil layers from the chronosequence data. What does this do to t , thus the integration interval, if you remove the upper layers from the data set? Additionally, is the model valid for sites receiving dust influx or is it only valid for sites with prolonged soil development in a single parent material? In some cases buried horizons were removed from the data

set. In case of a recent deposit on top of an older soil, is then the whole profile removed from the data set (section 2.3)? I guess not but how is the decision made?

4. Any non-linear or transient soil development, even when it is due to variations of EEMT (there may be other causes as well that the model does not consider, e.g. bio-turbation, frost action) is averaged out during the estimation of parameters of the bi-variate pdf by assuming it is constant. Transient soil changes occurring in the past may dominate todays' soil properties (e.g. ripening, erosion). I do not like statements like in line 149 that all processes are implicitly captured. There is no single process captured, only the net effect, made constant over time.

5. Is the soil forming factor "human activity" e.g. via tillage, fertilization, erosion inside or outside the scope of this model. If it is "implicit" as well, then the human influenced (time span of 100s of years) is smeared out over pedogenetic timescales, which is totally unrealistic. Making this point more general (see also point 3): what is assumed domain of validity? Natural soils receiving no dust influx and under fairly constant climate? The authors should comment on this in their discussion section.

6. Section 4.3 addresses quite a lot of the concerns that I make above and below, so the authors do realize these. The feeling remains, however, that the energy model approach because it is implicitly dealing with the mixture of soil forming processes occurring during the formation of the chronosequences, may not be able to deal with other mixtures in the future such as caused by human impacts and transient climate change.

Specific remarks: Line 94-97: If the model would be mechanistic, I would debate if the soil forming factors Climate+Organisms (Ps) are considered exchangeable with Re-lief+Parent material (Lo) (can they compensate each other). As it is empirical, I accept that the model is a (dark) gray box model with little representation of the processes but would not accept any mechanistic conclusions to be drawn with it.

Line 182-183: Depth weighted texture is in mass%. No account of mass distribution

over depth? In line 238 when converting to mass, a uniform bulk density of 1.5 (quite high) is assumed. As bulk density will vary over depths in many soils and is often related to (diagnostic) horizon occurrence, here we have yet another simplification.

Line 238-240: bulk density always 1500 kg m-3? Yet another simplification. If RF% is a needed input, to be measured on site, what is then the use of the estimation of the clay percentage on the same site (just analyze it, the pit has been dug). RF% is not cheaper to measure than clay%.

Section 3.1. L0 translates here as 3 parent materials. Can we say in general that due to the categorical nature of parent materials, equation 4 is dealing with the soil data in a stratified manner? To estimate the parameters of equation 6 in an unbi-ased manner, the strata (parent materials/rock types) should be sampled by probability sampling, which is obviously not the case as the input data are from chronosequence studies. The authors should add statements on the possible effect of biased sampling on estimated parameters.

Line 346: The statement on causes for clay content change are in contradiction with line 149 where a larger amount of processes are stated to be(implicitly) captured. As stated above, I think that the authors should not pretend with their approach to capture soil forming processes.

Section 4.1.1 mentions cases where the model does not predict well, which suggests again that the authors should define the model application range and confine the data domain for which it was fitted.

Line 439: It is not true that the strength of the approach is in the lack of assumptions about the initial conditions: parent material/rock type is used for stratification (eq.4).

Line 457: I do not believe that the parametrized approach makes any sense in pre-dicting potential landscape evolution. First, because the accelerating effect of human activities on soil formation and soil redistribution is not properly captured in the model,

second because the time integration is done using averaged EEMT. The authors should also state at what temporal extent they expect to be able to make landscape evolution forecasts. I think there could be additional value of this model to produce a covariate map in digital soil mapping that expresses the likely degree of weathering (via EEMT).

---

## Referee Comment (RC1) · J. Phillips (Referee) · 7 Nov 2016

Budiman Minasny has, correctly, called the ability to model soil formation and evolution the "holy grail" of pedology. Given the complex nonlinear dynamics and divergent development often observed in pedogenesis, along with issues of polygenesis and multiple causality (not to mention spatial variability and measurement uncertainty), a probabilistic approach to this problem makes sense. The title ("change through time") raised my hope that this paper would be a step in that direction. It is, but only a small step. The contributions of this paper are:

1. An improved correlation of some soil properties with climate and apparent age, based on total pedogenic energy (TPE).

[Figure]

2. A suggestion/demonstration that, as might be expected, local variations not captured by the soil-TPE correlation are closely related to landscape position and topography.

Though I suppose one could model change through time by varying the time factor for a given EEMT, this is not done in this paper and would be quite suspect due to the many assumptions and simplifications involved. The approach given here is, to be sure, probabilistic, but in a very traditional statistical uncertainty sense rather than in the more specific change-over-time insight I hoped for when I read the introduction.

I like the straightforward way this approach is linked to the classic factorial approach in lines 85-97. While the state factor model has its strengths and weaknesses, it is an appropriate framework for a model seeking applicability across a variety of environments and locations. The long chain of assumptions and simplifications are reasonable in the context of developing a relatively simple model, but move the final model more toward a black box and less a true factorial model.

Despite the simplifications and assumptions, and that the paper did not live up to my initial (perhaps too unreasonably high?) hopes, this is a worthwhile contribution to pedology, and a step toward true probabilistic modeling of pedogenesis.

Comments (some important, some trivial) on specific line numbers are below:

69-71: Could eliminate the first two "approaches."

128-131; 152-154: Why the bivariate normal density function? Were other pdf's considered?

144-154: These are some pretty serious and unrealistic assumptions.

159: "More than" rather than "over."

161-162: One of the "within the present study" phrases can be omitted.

164-166: Southern hemisphere and mid-continental sites are under-represented.

168-177: EEMT has been successfully used several times in the past decade, and is familiar to some pedologists. However, since the whole rationale of the method is based so heavily on S = f(TPE) = f(EEMT), and EEMT is not familiar to all pedologists, some additional explanation and background is called for.

182: I recommend framing this as examples of change through time rather than proxies, as many chemical and biological changes are not necessarily closely related to textural change.

213: Omit first "terrain."

232-236; 389-390: The influence of hillslope processes and morphology on soil development and thickness was recognized by the late 19th century. Soil depth does usually vary systematically across hillslopes, but these broad scale variations are often overlaid with complex local variations (see, e.g.. lines 335-6). Thus it seems a major stretch to claim that soil depth "corrects" for hillslope processes.

248-263: This section could use some clarification. The extent to which output from another model is used to both drive and evaluate the output of the probabilistic model is an issue. Perhaps a flow chart would help.

271 et seq.: Please clarify these correlations. Spearman correlations are not included in Table 2; text does not always make clear whether Pearson or Spearman is given.

322: Regression slope, presumably?

359-372: Also, sand and silt are dominantly resistant quartz, and present as primary minerals (or depositional inputs) rather than as a result of weathering or translocation. While weathering would be closely related to EEMT or TPE, residual primary minerals (or depositional inputs) would not be.

377-379: A number of references going back to at least the mid-1980s discuss both the dynamics of soil production and soil thickness, and associated nonlinearities (e.g., Johnson, Phillips, Minasny, D'odorico).

413: This implies some important scale influences worthy of further discussion.

421-424: Agreed. But in areas of complex lithology (most sedimentary rocks) this could be quite complicated.

426-462: I had hoped for something more explicitly probabilistic, such as predicting, e.g., ranges of clay content or the converging or diverging upper and lower limits over time. The approach given here is, to be sure, probabilistic, but in a very traditional scatter-around-trend sense rather than in the more specific change-over-time insight I hoped for when I read the introduction.

464-514: Good discussion of appropriate caveats.

467-470: Or even without controlling for other factors, especially in younger soils.

Table 2: Column headings are not labeled or identified in the caption. For Pearson correlation, why not use the traditional and familiar r, rather than p (or maybe it's supposed to be rho; hard to tell in this font) which normally denotes probability or confidence intervals?

---

## Referee Comment (RC2) · Anonymous Referee #2 · 8 Nov 2016

The manuscript entitled "A probabilistic approach to quantifying soil property change through time integration of energy and mass input" by Christopher Shepard et al. presents a probabilistic use of the energy and mass transfer model first developed by Runge in the seventies and modified more recently by Rasmussen and co-authors. This use implies an adaptation of the energy and mass transfer model (EEMT) by including the soil formation duration within the model. The proposed adaptation of the EEMT model is then applied to three data set of different natures: a large set of chronosequences gathered from the literature; a compilation from upland catchment from the US NSF Critical Zone Observatory, and a small complex catchment in the Santa Catalina Mountains. The application of the model to this last dataset requires

further modification of the initial model. As such, this work present an interesting contribution that is worth publishing. However, the objectives of the proposed model are unclear to me. Indeed modelling of soil may have different scopes: the understanding of soil formation, the prediction of soil evolution in future or the prediction of the soil distribution in space. Contrarily to what is stated in the title, introduction and/or discussion, the proposed model cannot be used to either understand the soil formation or to predict its evolution in future as it is not a mechanistic model. Indeed the understanding of the soil formation now a days requires the understanding of the interactions and retroactions among soil processes. Only an understanding may bring some gain of knowledge on the threshold and chaotic aspects of soil formation. Concerning the prediction of the evolution of soils within future, the proposed model has to integrate both the climate evolution and the human activity. As far as I understood, climate is considered as constant (lines 144-145) in the proposed model and human activity is not considered at all. The proposed model can therefore mainly be used to represent spatially the distribution of some soil properties providing that the age of the soil is known. This last data is not easily to obtained and limit thus strongly the applicability of the model. Considering this, the title, the introduction and the discussion of the paper should be modified. Since the paper is based on the EEMT model, this model should be at least shortly presented to facilitate the understanding of the paper for all readers (including those who have not read the papers by Runge and Rasmussen). In addition, the data set should be better described providing the age span of the studied soils, their depth range, stoniness, the vegetation.... Indeed the relative variability over the data sets of one parameter compared to the other may explain their relative importance in the response of the model to the different parameters e.g. paragraph from line 319 to line 326. It would be interesting to present a repartition of the number of soil studies per parent material, vegetation, climate types in order to get a better idea of the representatively of the studied soils compared to the worldwide variety and thus of the applicability of the model. In paragraph from line 359 to line 372, the authors show that the clay content is well predicted by the model while the silt and

sand fractions are not, the worst result being for the sand fraction. This result seems expectable to me as the sand fraction contains a large variety of primary minerals that strongly vary according to the geology of the parent material and have strongly variable ability to weather. Therefore, the bad prediction of the sand fraction may be related to the impact of the initial parent material condition on the model. Such an impact is also observed in the bad prediction of the clay content for the soils developed on amphibolite (Fig. 7a). Therefore it is not true to conclude that the model is not sensitive to the initial conditions. This statement should be minored in the revised version of the manuscript.

Please also note the supplement to this comment:
http://www.soil-discuss.net/soil-2016-63/soil-2016-63-RC2-supplement.pdf

---

## Author Comment (AC1) · 17 Jan 2017

Response to Editor Peter Finke

Shepard C., Schaap MG., Pelletier JD., and Rasmussen C.

We thank the editor for his comments and recommendations on the manuscript titled "A probabilistic approach to quantifying soil property change through time integration of mass and energy input." We have responded to and addressed the editor's comments and remarks below and in the revised version of the manuscript.

Response to general remarks: The model requires the inputs of time and effective energy and mass transfer (EEMT, Rasmussen et al., 2005; Rasmussen and Tabor,

2007; Rasmussen et al., 2011) to predict the probable range of a particular soil physical property at a given location. In this presentation, we focus on clay content as a physical property that reflects pedogenic change. EEMT is clearly rooted in the classical Jenny factorial approach for describing soil forming system, and as such does not describe any particular soil forming process. EEMT is a flux term, and is it not used here to parameterize a mechanistic model of pedogenesis as suggested by the editor. EEMT simply quantifies the energetic contributions from effective precipitation and net primary productivity as quantitative measures of climatic and biological forcing/input to the soil forming system. EEMT quantifies the energy transferred to the soil system that can perform pedogenic work, such as chemical weathering or carbon cycling, or any other soil forming process; it does not describe or quantify any one process, and this is indicated on lines 102-105.

Application of the present approach in the critical zone environment requires no soil information. Here we used an established geomorphically based numerical model that predicts local erosion rates, soil depth and local soil residence time from topography and a maximum rate of soil production (that can be assumed or based on local catchment derived denudation rates) (Pelletier and Rasmussen, 2009). The geomorphic model is mechanistic and process based, describing mass production and transport using established transport "laws". The editor expressed doubt of the ability to predict soil information in the critical zone environment; however, we present clear model results that this approach can be used to predict clay content completely independent of soil data. This is a key piece of soil information to understanding critical zone function and evolution. We further argued that the present approach can greatly inform our understanding of the distributions of soil physical properties and facilitate further hypothesis generation. For example, the present approach did not accurately predict clay stocks at specific locations within the Santa Catalina Mountains-Jemez River Basin granite sub-catchment (Lines 329-339); any number of hypotheses and questions can be formulated and tested as to why the model failed to predict clay stocks at these locations, and the current model formulation can be updated to accommodate

these findings. Further, the present results suggest an incomplete understanding of the soil-landscapes within these catchments, which may not have been found by using techniques such as digital soil mapping.

The strength of the current approach lies in the ability of the model to capture all soil forming factors into one relatively simple mathematical apparatus. We make no claims of modeling particular soil forming processes, a fact that we state clearly in lines 145-149 and in lines 151-152. As true of any factorial treatment of soil systems, the model captures either the net effect of all considered soil forming processes or rather the implicit result of soil forming processes, by considering all soil profiles equally. This is the same foundation for any number of digital soil mapping exercises, as typified by the SCORPAN statement of McBratney and others – the model uses factors to predict soil properties and is not a mechanistic model of process. The model only indirectly captures soil forming processes by not restricting the model to any particular spatial or temporal extent or any particular parent material. We disagree with the editor's comment about the delineation of the model domain. By restricting the model domain, either spatially, temporally, or with regards to parent material, entire suites of soil forming processes would not be captured, limiting the applicability of the present approach.

Response to Question 1. Why is the model forced to a bivariate pdf form? Techniques were on the shelf?

A bivariate normal density distribution was used for the present approach because it generally represents the mathematically simplest bivariate distribution and is easily parameterized. However, we did not consider other bivariate density functions, we wanted to demonstrate proof of concept before exploring complexities or refinements to the approach. We have added language to the revised manuscript at lines 131, indicating that we choose to force the data to a bivariate normal structure and that other density functions are available and may provide a better fit to the soil physical property of interest.

Response to Question 2. Question with regards to quantifying soil age and EEMT.

The editor is correct in that the model assumes EEMT is constant over the duration of pedogenesis. Further, we agree with the reviewer that climate throughout the Quaternary has not been constant (Zachos et al., 2001), and that this inconsistency likely has influenced our predicted soil property values. We directly addressed this model limitation in section 4.3, lines 503-514:

"Furthermore, global climate patterns have shifted dramatically over the last 65 Mya (Zachos et al., 2001). The majority of soils observed in the compiled chronosequence database span the Quaternary, including both the Holocene and Pleistocene. The Pleistocene was marked by a number of major glacial-interglacial cycles at approximately 100,000-year intervals (Imbrie et al., 1992; Wallace and Hobbs, 2006), which corresponded with shifting climatic conditions, e.g., for large portions of the northern mid-latitudes glacial periods were generally cooler and wetter, and interglacial periods were warmer and drier (Connin et al., 1998; Petit et al., 1999). Further, the Pleistocene climate shifts likely influenced the rates of weathering and clay production (Hotchkiss et al., 2000). Taking into account the differences in past and modern climate would likely diminish disparities between observed and modeled soil physical properties. Reconstructed global paleo-EEMT values would improve model accuracy, and limit uncertainty in the probabilistic ranges of soil properties for soils older than Holocene age."

Further, we discussed a possible model correction in which paleo-EEMT values could be calculated and used to provide better estimates of TPE. However, in order to calculate an accurate accounting of paleo-EEMT values would require datasets of about past mean monthly air temperature, mean monthly precipitation and monthly net primary productivity (Rasmussen et al., 2005, 2011; Rasmussen and Tabor, 2007) for the entire duration of pedogenesis. Unfortunately, few, if any, locations on the planet have spatially explicit paleoclimatic records with all the necessary data requirements to perform this calculation, although paleoclimate predictions are improving, e.g. the recent CIMP4 general circulation model application to predict global LGM climates, which
represents an ongoing opportunity to incorporate such data into pedogenic models. As such, we made the simplifying assumption that the current climate can be used to represent of climates that many of the included soils evolved under. This is true of any factorial approach and representation of soil data that includes soils older than the Holocene. Any representation of soil properties relative to mean annual temperature or precipitation or any plot of soil property change vs time invariably includes past climate variation influence of soil property evolution. We clearly stated and recognized this in the text.

Defining soil age is a challenge in many landscapes as the editor suggests; however, there are simple techniques that can be used to estimate soil age without the need for expensive cosmogenic radionuclide dating. The age of geomorphic landforms can be estimated by using the cross-sectional shape of gully cuts or scarp-like surfaces and hillslopes and a known hillslope diffusivity value (Bucknam and Anderson, 1979; Hsu and Pelletier, 2004; Pelletier et al., 2006; Pelletier and Cline, 2007). Estimating geomorphic landform age requires only the use of either a digital elevation model (DEM), or profiles of scarp elevation, both of which are easily and inexpensively attained. Further, relative age dating is widely used in chronosequence studies and provides general estimates and constraints of soil-geomorphic surface ages (Schaetzl and Anderson, 2005). With regards to upland catchments, catchment averaged denudation rates can be estimated from cosmogenic radionuclides (CRN) using a smaller number of samples than would be necessary for quantifying full CRN depth profiles (Granger et al., 1996). Using a geomorphically based model of soil depth, spatially explicit soil ages can be calculated was discussed in lines 220-224. As such, we do not agree with the editor the target variables are more easily determined than soil age; for any chronosequence study, soil age would have to be determined regardless of the target variables of interest, assuming soil age is too expensive or indeterminable is not an appropriate or accurate critique of the presented model.

Response to Question 3. Removal of buried horizons.

Buried horizons were removed from the dataset, as we assumed that these buried horizons were not reflective of the relationship between the modern climate and the subaerial soil horizons. We decided to remove buried horizons, as the subaerial soil horizons likely are more correlated with the current climate, as compared to the buried horizons. Eolian horizons were removed from soils described in McFadden et al. (1986), because these horizons are likely significantly younger than the basaltic flow dates that were used to represent the soil age; however, based on the reviewer's suggestion we have removed these soils from the current dataset and updated lines 194 to reflect this change. Only buried horizon have been removed from the dataset presented in the revised manuscript.

Response to Question 4. Non-linear or transient soil formation.

We specifically addressed non-linear soil formation and internal or intrinsic feedbacks driving soil development in section 4.3, lines 490-502: "Internal or intrinsic feedbacks and thresholds within the soil system drive pedogenic development without changes in the external state factors (Muhs, 1984; Chadwick and Chorover, 2001). For example, greater chemical weathering and clay production due to increased water residence time caused by argillic horizon development is the result of an internal feedback that is independent of the external climatic and biological system (Schaetzl and Anderson, 2005). These thresholds can operate as progressive or regressive processes, driving soil formation forward or hindering further development (Johnson and Watson-Stegner, 1987; Phillips, 1993). Internal soil development feedbacks were not explicitly considered in the present model formulation. The presence of these internal feedbacks may partially explain error within the model predictions. Changes in EEMT would not explain all observed differences in soil properties over the age of the soil. However, if these feedbacks were operating in the included soils, the influence of intrinsic thresholds was implicitly captured within the probability distributions, partially accounting for the role of internal soil development feedbacks on soil formation."

The model does capture all soil forming processes implicitly, in that no one process

is explicit expressed or quantified. Further, we agree with the reviewer that the model does produce a prediction of soil physical properties based on the net effect of these soil forming processes. We have edited lines 149-153 by removing the word "implicitly".

Response to Question 5. Human impacts on soil formation.

We did not address human impacts within the current manuscript or even discuss anthropogenically driven changes in land use or climate. Here we demonstrate the use of a probabilistic model for quantifying the distribution of soil properties we observe currently on the Earth's surface that has arisen during the Quaternary. Furthermore, the energetic contributions from human impacts and dust influx can and have been incorporated within the EEMT apparatus (Rasmussen et al., 2011). The energetic inputs from dust or fertilizer additions, for example, are generally orders of magnitude smaller than the energetic inputs from solar radiation, precipitation, or primary productivity into the soil system. The energetic inputs to the soil from other direct human activities such as the compression of soil due to farming equipment or the increased erosion due to construction or plowing (Rasmussen et al., 2011). We have added language to indicate the adaptability of the EEMT model for differing soil environments at lines 212-215 in the revised manuscript. In specific systems, both dust inputs and human impacts may be significant, however, the vast majority of the soils included in the presented dataset are not directly impacted by human activities or modern dust influx. As the model is probabilistic in nature, the model can simply predict a probable range of target soil physical properties, the domain is generally unconfined. As stated above, human and dust inputs to the soil system can be incorporated into EEMT allowing the inclusion of these soils within the model. Furthermore, the application of the model in this manuscript was to predict clay content, this is a soil property that does not readily change over human time-scales – but rather reflects geologic time scale pedogenic change.

Response to Question 6. Regards to use of energetic models.

Energetic approaches to quantifying soil physical properties and soil formation are able to deal with differing mixtures of the soil forming factors. The soil state factor model has the potential to be expanded beyond the classical five soil forming factors to include influences from site-specific soil forming factors such as the addition of fertilizer to soils or increased erosion due to human activity (Jenny, 1941, 1961). All energetic pedogenic models are derived from the soil state factor model (Minasny et al., 2008), and as such have the potential to be expanded to accommodate additional soil forming factors. Energetic approaches account for the potential fluxes of matter and energy into the soil system that are associated with the soil forming factors, relating the energetics of the fluxes into the soil system to soil physical properties and structures (Volobuyev, 1964; Runge, 1973; Smeck et al., 1983; Rasmussen et al., 2005, 2011; Rasmussen and Tabor, 2007). The energetic input from fertilizer can be easily quantified and included with the model scheme for EEMT when appropriate, however, the energetic additions to the soil from fertilizer are orders of magnitude smaller when compared to the energetic additions of soil radiation, effective precipitation, and net primary productivity (Rasmussen et al., 2011).

The impact of anthropogenic climate change on soil physical properties can be incorporated into the EEMT model space. EEMT can be updated to include the impacts of increased atmospheric CO2 on the fluxes of matter and to the soil system. Local changes in air temperature, precipitation, evapotranspiration, and net primary productivity due to increased atmospheric CO2 are quantifiable, and can be easily incorporated into the EEMT model. Further, EEMT can be calculated on a range of temporal scales from near-real time through the use of eddy flux tower and meteorological flux measurements to annually (Rasmussen et al., 2015). With EEMT values updated to include the impacts of anthropogenic climate change, the presented model structure is capable of incorporating these influences on soil formation. As such, we disagree with the reviewer that energetic-based pedogenic models are not capable of handling changing earth surface conditions, and can be updated to accommodate human influences on the climate and landscapes.

Response to specific remarks

Response to remarks on Lines 94-97, interchangeability of Px and Lo:

The soil state factor model developed by Jenny (S=f(cl,o,r,p,t)) was later formulated by Jenny as S=f(Px,Lo,t), recognizing that climate and biology are generally flux inputs into the soil system and relief and parent material are site factors. Px influences pedogenesis and soil evolution over the lifetime of the soil and may be time dependent, whereas Lo generally represents the initial state of the soil forming system and is not time dependent (Jenny, 1961). Interchanging the influence of Px and Lo is not possible. Relief or topography can vary over time, and in certain formulations of the soil state factor model may be considered time dependent, however, the chronosequence data used to parameterize the present approach are sited on low sloping surfaces, and changes in topography were minimized. Furthermore, as described by Jenny, and approximately 70 decades of soil science, the soil state factors do not describe soil forming processes, the state factors only describe the soil forming environment, i.e. climate is not a soil forming process, but a description of the conditions under which certain soil properties are observed and soil processes operate.

Response to remarks on Lines 182-183: Depth weighted percent clay calculation and bulk density values

We used depth weighted average percent clay in the prediction of clay stocks to account for the greater influence of thicker soil horizons on the account of clay stocks. By calculating a depth weighted average we are accounting for the distribution clay with depth, and summarizing those values into one value. Our model was trained on depth weighted average clay percentages from the chronosequence database; consequently, we also used depth weighted average clay values for predicting clay on a mass per area basis. We agree with the editor, bulk density is not constant throughout a profile, unfortunately, bulk density is difficult to measure, or is often not measured in the field. Further, bulk density data are not commonly reported within the soil science

literature or in the available chronosequence literature. Without the necessary data, we chose to assume a constant value of 1500 kg m-3 for all soil profiles used in the calculation of predicted mass per area clay. If bulk density data were available, those data could be easily included in the prediction of mass per area, and likely the presented probabilistic-energetic would likely better predict clay stocks.

Response to remarks on Lines 238-240: Assumption of 1500 kg m-3 bulk density and use of RF% for calculating clay stocks.

Bulk density is not a commonly measured soil variable as it is often difficult to obtain measurements for bulk density from soil profiles, and values for bulk density are highly method dependent; there was low reporting of bulk density estimate in the available chronosequence literature. Due to a lack of measured values, a constant value was chosen for all profiles; if bulk density measurements are available than the measured values should be used in the predictions of clay stocks. Further, RF% data were used in the predicted clay stocks, as (1-RF%) in Eq. 9 describes the volume or fraction of the soil profile in which clay sized particles accumulate. Additionally, RF% did not influence the prediction of clay stocks (line 326); if RF% data were unavailable, a standard or constant value could be assumed for predicting clay stocks. Further, these simplifications for calculating soil properties on a mass per area basis are standard corrections and assumptions that are made throughout the available literature. With missing or incomplete data, the complexities of measuring soil properties in the field, educated assumptions are usually required.

Response to remarks on Section 3.1 Bias in sampling and stratification of Lo:

The editor did not fully understand the model background as presented in the manuscript. Lo was not used for stratification; Lo is not directly expressed within the model structure. We removed Lo from Eq. 4 to produce Eq. 5, justifying this simplification as time partially accounts for the influence of topography and parent material variation. Soil residence time on a landscape is proportional to slope or curvature

(Heimsath et al., 1997, 2002; Yoo et al., 2007). Additionally, the degree of weathering or alteration of the parent material and the presence of secondary minerals and products are also proportional to the soil residence time (Brimhall and Dietrich, 1987; Chadwick et al., 1990; Brimhall et al., 1992). We chose to break down model predictions using leave one out cross validation by parent material in Fig 4 to demonstrate the model was insensitive to different parent materials or landforms, the predictive ability of the model did not vary significantly between the 3 broad parent material categories. We did not calculate model parameters based on parent material, we presented global parameter values in Table 2. We have updated lines 124 and lines 155-157 to clarify this point.

Biased sampling due the use of chronosequence studies is an issue faced in all of soil science. Soil pits are generally preferentially sited in locations where it is possible to dig a soil pit of sufficient depth to sample the soil profile. Any chronosequence study or synthesis of chronosequence data is hampered by biases within soil sampling and presentation of selected data in the literature. Biases in estimated model parameters are based upon sampling techniques and availability of chronosequence data in the literature not due to selective sampling of chronosequence data used to calculate model parameters. We did not limit the data used to calculate the model parameters from the chronosequence literature as a way to minimize errors within the presented model.

Response to remarks on Line 346, clay content change:

In lines 151-152 we stated: "The model assumes all changes in soil physical properties are due to pedogenic processes.", and in lines 345-348 we stated: "Weathering and clay production are primary pedogenic processes (Birkeland, 1999; Schaetzl and Anderson, 2005), and because the model assumed all changes in the soil profile are due to these processes, the model was the most effective at predicting clay content." These statements are in agreement with each other. The model implicitly captures the net effect of all pedogenic processes, we assumed that all changes in the soil profiles are due to pedogenic processes, and the primary pedogenic processes are weathering and clay production, amongst a suite of other processes (Schaetzl and Anderson, 2005), meaning the model captures the influence of weathering and clay production in soil property change. There is no "pretending", we are not misrepresenting the model or its predictive power in any way. We did not claim to model soil forming process, only the end result of all soil forming processes at specific times and energetic inputs based on the available chronosequence literature; this is true for any chronosequence or time based representation of soil data that has been published over the last 70+ years.

Response to remarks on Section 4.1.1, and where the model underperformed:

In many areas, estimated TPE values likely do not account for the total flux of mass and energy into the soil system. Error in predicted clay percentages are likely partially a function of error in TPE estimates. We discussed underestimations of TPE due to changing climate, topography, parent material differences and intrinsic thresholds in soil formation in Section 4.3. Further, we did not constrain the chronosequence data set used to calculate the model parameters, as the formulated model is capable of handling soil data from a wide range of environments and locations due to its probabilistic component. We highlighted where the model failed to predict soil property values as a way to highlight locations where we still have an incomplete understanding of soil formation, or places where parent material greatly influences resultant soil formation (i.e. coral reef terraces). The inability of the model to predict soil properties in particular areas, suggests that a soil-forming factor not included in EEMT or TPE is highly influencing soil formation in this area, not that input data used to calculate model parameters need to be constrained. Models are only representations of reality and there is no logical need for a model to perform perfectly. It is generally beneficial to identify locations and conditions under which models do not work, as a way to identify potential model refinements.

We did not highlight model failures as an excuse to selectively choose data to achieve the best model predictions as suggested by the editor. Further, constraining data to achieve a successful model prediction is uninteresting, as one cannot identify locations

or conditions under which the model breaks. Without the inclusion of coral reef terrace chronosequence, we would not have identified that the model has an inability to predict resultant soil formation under fine textured parent materials and tropical climates.

Response to remarks on Line 439, model assumptions about initial conditions:

The editor did not understand the model background as written in lines 115-127, we did not stratify the data by parent material:

"Explicitly including time in Eq. 4 through TPE partially captures variation in soil property change attributable to topography and parent material. Soil residence time may be directly related to landscape position through topographic control on soil production and sediment transport/deposition (Heimsath et al., 1997, 2002; Yoo et al., 2007). Additionally, parent material modulates soil residence time through control on soil depth (Rasmussen et al., 2005; Heckman and Rasmussen, 2011), soil production, and sediment transport rates (Andre and Anderson, 1961; Portenga and Bierman, 2011). The initial conditions of the soil forming system ($L_o$) are never fully known; however, representing the state of the soil system as a probable distribution of values, implicitly accounting for soil age, and not constraining the initial soil forming conditions, Eq. 4 can be stated simply as: $P(s\_1{\leq}S{\leq}s\_2)=f(TPE)$ (5) where the probability state of the soil, $P(s\_1{\leq}S{\leq}s\_2)$, bounded by a lower and upper limit, is a function of one quantifiable variable."

We simply removed $L_o$ from Eq. 4 to write Eq. 5, TPE partially accounts for variation in $L_o$ due to the influence of topography and parent material on soil residence time, as discussed above. We did not use parent material to stratify model parameter estimates; we calculated model parameters for the entire chronosequence dataset. We did not make assumptions about the soil parent material, or include any data about the parent material within the presented model. The statement in Line 439 is accurate. We have updated lines 124 and lines 155-157 to clarify this point.

Response to remarks on Line 459, potential to model landscape evolution:

We strongly disagree with this statement. First, the application of the model that couples the geomorphic model of Pelletier and Rasmussen (2009) explicitly includes soil production and sediment transport to predict landscape variation in soil depth and residence time – these values coupled with TPE yield estimates of soil physical properties completely independent of any soil data. This coupled model may be used to predict soil and landscape evolution across any range of topographic and/or climate scenarios, and yield probabilistic estimates of soil clay content. As stated throughout the manuscript, the model was designed to capture Quaternary soil evolution; additionally, the focus on clay content necessitates a geologic time scale perspective as this property changes not on human scales, but on pedogenic time scales. Changes induced by human activity could be incorporated into the sediment transport of the Pelletier and Rasmussen (2009) model, as well as incorporated into the energy and mas transfer terms, but in terms of changes in clay content over human time scales these changes will likely be insignificant. As stated in the manuscript, this approach could be used to investigate potential landscape scenarios. Using the geomorphic model (Pelletier and Rasmussen, 2009), potential landscape evolution scenarios could be investigated where changes in topography and soil thickness are used to determine changes in soil properties across small watersheds. We stress that potential landscape evolution scenarios could be investigated, assuming the landscape is at steady state, soil development and evolution could be teased apart using the presented approach; any predictions drawn from such hypothetical modeling exercises would only be a potential future for any landscape. Furthermore, as discussed in this review EEMT can be updated to include the influence from human impacts on the atmosphere and landscapes, and TPE can also be calculated to include the influence of a changing climate (Rasmussen et al., 2011).

As stated in the manuscript and repeatedly above, we used modern EEMT integrated over the age of the soil as the estimate of "total" pedogenic energy input as the best available data that we have. We clearly recognize that this does not incorporate past climate change, leading to what could be under/over estimates of TPE depending on how

the local climate system changed at each included location during glacial periods. The majority of sites were from northern hemisphere mid-latitude sites suggesting modern EEMT, and hence, TPE likely underestimates the total pedogenic energy transferred to each location. As noted spatially explicit estimates of LGM climate conditions are now available, but we currently lack a time resolved estimate of paleoclimate variation. As such, we used modern values as proxies for soil forming factors. This is true for any study of soil properties relative to modern climate. Based on the editor suggestion we have removed the reference to investigation of potential soil forming environments at lines 540-541 in the revised manuscript.

References Andre, J., and H. Anderson. 1961. Variation of Soil Erodibility with Geology, Geographic Zone, Elevation, and Vegetation Type in Northern California Wildlands. J. Geophys. Res. 66(10): 8.

Birkeland, P.W. 1999. Soils and Geomorphology. Third. Oxford University Press, New York, New York.

Brimhall, G.H., O. a Chadwick, C.J. Lewis, W. Compston, I.S. Williams, K.J. Danti, W.E. Dietrich, M.E. Power, D. Hendricks, and J. Bratt. 1992. Deformational mass transport and invasive processes in soil evolution. Science 255(5045): 695–702Available at http://www.sciencemag.org/content/255/5045/695.short.

Brimhall, G.H., and W.E. Dietrich. 1987. Constitutive mass balance relations between chemical composition, volume, density, porosity, and strain in metasomatic hydrochemical systems: Results on weathering and pedogenesis. Geochim. Cosmochim. Acta 51: 567–587.

Bucknam, R.C., and R.E. Anderson. 1979. Estimation of fault-scarp ages from a scarp-height - slope-angle relationship. Geology 7: 11–14.

Chadwick, O.A., G.H. Brimhall, and D.M. Hendricks. 1990. From a black to a gray box — a mass balance interpretation of pedogenesis. Geomorphology 3(3-4): 369–

390Available at http://linkinghub.elsevier.com/retrieve/pii/0169555X9090012F.

Chadwick, O.A., and J. Chorover. 2001. The chemistry of pedogenic thresholds. Geoderma 100(3-4): 321–353Available at http://linkinghub.elsevier.com/retrieve/pii/S0016706101000271.

Connin, S., J. Betancourt, and J. Quade. 1998. Late Pleistocene C4 plant dominance and summer rainfall in the southwestern United States from isotopic study of herbivore teeth. Quat. Res. 50: 179–193Available at http://www.sciencedirect.com/science/article/pii/S003358949891986X (verified 15 February 2015).

Granger, D.E., J.W. Kirchner, and R. Finkel. 1996. Spatially averaged long-term erosion rates measured from in situ-produced cosmogenic nuclides in alluvial sediment. J. Geol. 104(3): 249–257.

Heckman, K., and C. Rasmussen. 2011. Lithologic controls on regolith weathering and mass flux in forested ecosystems of the southwestern USA. Geoderma 164(3-4): 99–111Available at http://linkinghub.elsevier.com/retrieve/pii/S0016706111001133 (verified 4 February 2015).

Heimsath, A.M., J. Chappell, N.A. Spooner, and D.G. Questiaux. 2002. Creeping soil. Geology 30(2): 111Available at http://geology.gsapubs.org/cgi/doi/10.1130/0091-7613(2002)030<0111:CS>2.0.CO;2.

Heimsath, A.M., W.E. Dietrich, K. Nishiizumi, and R.C. Finkel. 1997. The soil production function and landscape equilibrium. Nature 388(July): 358–361.

Hotchkiss, S., P.M. Vitousek, O.A. Chadwick, and J. Price. 2000. Climate Cycles, Geomorphological Change, and the Interpretation of Soil and Ecosystem Development. Ecosystems 3(6): 522–533Available at http://link.springer.com/10.1007/s100210000046 (verified 21 October 2014).

Hsu, L., and J.D. Pelletier. 2004. Correlation and dating of Quaternary alluvial-fan

surfaces using scarp diffusion. Geomorphology 60(3-4): 319–335.

Imbrie, J., I.E.A. Boyle, S.C. Clemens, A. Duffy, I.W.R. Howard, G. Kukla, J. Kutzbach, D.G. Martinson, A. McIntyre, A.C. Mix, B. Molfino, J.J. Morley, N.G. Pisias, W.L. Prell, L.C. Peterson, and J.R. Toggweiler. 1992. On the structure and origin of major glaciation cycles 1. Linear responses to Milankovith forcing. Paleoceanography 7(6): 701–738.

Jenny, H. 1941. Factors of Soil Formation: A System of Quantitative Pedology. Dover Publications, Inc, New York, New York.

Jenny, H. 1961. Derivation of state factor equations of soils and ecosystems. Soil Sci. Soc. Am. J.: 385–388Available at https://dl.sciencesocieties.org/publications/sssaj/abstracts/25/5/SS0250050385 (verified 29 January 2015).

Johnson, D., and D. Watson-Stegner. 1987. Evolution model of pedogenesis. Soil Sci. 143(5): 349–366Available at http://journals.lww.com/soilsci/Abstract/1987/05000/Evolution_Model_of_Pedogenesis.5.aspx (verified 6 November 2014).

McFadden, L.D., S.G. Wells, J.C. Dohrenwend, and M. Park. 1986. Influences of Quaternary climatic changes on processes of soil development on desert loess deposits of the Cima Volcanic Field, California. Catena 13: 361–389.

Minasny, B., A.B. McBratney, and S. Salvador-Blanes. 2008. Quantitative models for pedogenesis — A review. Geoderma 144(1-2): 140–157Available at http://linkinghub.elsevier.com/retrieve/pii/S0016706107003692 (verified 30 August 2014).

Muhs, D.R. 1984. Intrinsic thresholds in soil systems. Phys. Geogr. 5: 99–110. Pelletier, J.D., and M.L. Cline. 2007. Nonlinear slope-dependent sediment transport in cinder cone evolution. Geology 35(12): 1067–1070.

Pelletier, J.D., S.B. DeLong, a. H. Al-Suwaidi, M. Cline, Y. Lewis, J.L. Psillas, and B. Yanites. 2006. Evolution of the Bonneville shoreline scarp in west-central Utah: Comparison of scarp-analysis methods and implications for the diffusions model of hillslope evolution. Geomorphology 74(1-4): 257–270.

Pelletier, J.D., and C. Rasmussen. 2009. Geomorphically based predictive mapping of soil thickness in upland watersheds. Water Resour. Res. 45(9): n/a–n/aAvailable at http://doi.wiley.com/10.1029/2008WR007319 (verified 21 October 2014).

Petit, J., J. Jouzel, D. Raynaud, and N. Barkov. 1999. Climate and atmospheric history of the past 420,000 years from the Vostok ice core, Antarctica. Nature 399: 429–436Available at http://www.nature.com/articles/20859 (verified 15 February 2015).

Phillips, J.D. 1993. Progressive and Regressive Pedogenesis and Complex Soil Evolution. Quat. Res. 40: 169–176Available at http://www.sciencedirect.com/science/article/pii/S0033589483710690 (verified 6 November 2014).

Portenga, E.W., and P.R. Bierman. 2011. Understanding earth's eroding surface with 10Be. GSA Today 21(8): 4–10.

Rasmussen, C., J.D. Pelletier, P.A. Troch, T.L. Swetnam, and J. Chorover. 2015. Quantifying Topographic and Vegetation Effects on the Transfer of Energy and Mass to the Critical Zone. Vadose Zo. J.Available at https://dl.sciencesocieties.org/publications/vzj/abstracts/0/0/vzj2014.07.0102.

Rasmussen, C., R.J. Southard, and W.R. Horwath. 2005. Modeling Energy Inputs to Predict Pedogenic Environments Using Regional Environmental Databases. Soil Sci. Soc. Am. J. 69(4): 1266–1274Available at https://www.soils.org/publications/sssaj/abstracts/69/4/1266 (verified 3 November 2014).

Rasmussen, C., and N.J. Tabor. 2007. Applying a Quantitative Pedogenic Energy

Model across a Range of Environmental Gradients. Soil Sci. Soc. Am. J. 71(6): 1719Available at https://www.soils.org/publications/sssaj/abstracts/71/6/1719 (verified 27 October 2014).

Rasmussen, C., P.A. Troch, J. Chorover, P. Brooks, J. Pelletier, and T.E. Huxman. 2011. An open system framework for integrating critical zone structure and function. Biogeochemistry 102(1-3): 15–29Available at http://link.springer.com/10.1007/s10533-010-9476-8 (verified 21 October 2014).

Runge, E.C.A. 1973. Soil Development Sequences and Energy Models. Soil Sci. 115(3): 183–193.

Schaetzl, R., and S. Anderson. 2005. Soils: Genesis and Geomorphology. First. Cambridge University Press, Cambridge, UK.

Smeck, N., E. Runge, and E. Mackintosh. 1983. Dynamics and genetic modelling of soil systems. p. 51–81. In Wilding, L., Smeck, N., Hall, G. (eds.), Pedogenesis and Soil Taxonomy I. Concepts and Interactions. Elsevier, Amsterdam, ND.

Volobuyev, V. 1964. Ecology of soils. Academy of Sciences of the Azerbaijan SSR. Institute of Soil Science and Agronomy. Israel Program for Scientific Translations., Jerusalem, Israel.

Wallace, J.M., and P. V Hobbs. 2006. Atmospheric Science: An Introductory Survey. Second. Academic Press Inc., Amsterdam, ND.

Yoo, K., R. Amundson, A.M. Heimsath, W.E. Dietrich, and G.H. Brimhall. 2007. Integration of geochemical mass balance with sediment transport to calculate rates of soil chemical weathering and transport on hillslopes. J. Geophys. Res. F Earth Surf. 112(2): F02013Available at http://doi.wiley.com/10.1029/2005JF000402.

Zachos, J., M. Pagani, L. Sloan, E. Thomas, and K. Billups. 2001. Trends, rhythms, and aberrations in global climate 65 Ma to present. Science (80-. ). 292(April): 686–694Available at http://www.sciencemag.org/content/292/5517/686.short (verified

14 February 2015).

---

## Author Comment (AC3) · 17 Jan 2017

Response to Anonymous Reviewer #2

We thank the reviewer for their comments on the manuscript titled: "A probabilistic approach to quantifying soil property change through time integration of energy and mass input". Below we have detailed our response to the reviewer's comments, including how the manuscript was edited.

We disagree with a number of the assertions made by the reviewer. The objective of the presented work was to present the development of a probabilistic model of soil property change using time integrated energy and mass inputs. We added language

clarifying this hypothesis at lines 78-80 in the revised manuscript. We think the title is appropriate for the manuscript as written; the manuscript discusses modeling the change in soil properties using energy and mass inputs integrated with time or the age of the soil system, as such we did not change the manuscript title. Further, we did not attempt to model any soil forming process. The presented probabilistic approach is based upon the soil state factor model, which only considers the state of the soil forming environment to understand soil formation or soil structures (Jenny, 1941, 1961). Similarly, the effective energy and mass transfer model only quantifies the amount of potential heat and chemical energy added to the soil, it does not describe any soil forming processes (Rasmussen et al., 2005, 2011, 2015; Rasmussen and Tabor, 2007). This is clearly stated in lines 85-87 and 145-152 in the original manuscript. The model simply quantifies the net effect of all soil forming processes operating within the soil forming system. The revised manuscript has been edited to clarify this point at lines 154. Further, the discussion focused on the reasons for poor or acceptable model fits for the modeled soil properties, and discusses advantages and disadvantages to the probabilistic approach. Nowhere within the discussion do we claim that the presented approach models any soil forming process or soil formation. As was stated in lines 145-152 of the original manuscript, the model assumes all changes within the soil are due to pedogenic processes, this aligns with what is stated in the discussion in lines 345-348 in the original manuscript. EEMT and TPE quantify the amount of energy added to the soil system capable of doing pedogenic work, e.g. chemical weathering, which is most representative of clay formation.

The influence of climatic variability on the model results is discussed in length in lines 503-514 in the original manuscript. While paleoclimatic reconstructions are available, such as the spatially explicit LGM paleoclimate reconstruction from the CIMP4 general circulation model, time resolved paleoclimatic reconstructions are still largely unavailable for many locations. We chose to use modern climate values as they represent the best available data. We did not discuss human or anthropogenic impacts on soil formation or evolution. We are exclusively focused on Quaternary soil formation. In

this manuscript, we focused on clay formation, which occurs on a geologic timescale and does not readily change on human timescales. Additionally, the EEMT model is easily adaptable to accommodate human influences on the energy and mass inputs into soil, such as fertilizer inputs (Rasmussen et al., 2011).

We have added the age span and depth ranges to the revised manuscript at lines 179-180. Data on vegetation and stoniness were not available across all the sequences and were not included. Further, many of these data are included in Table S1, where all of the chronosequences are listed along with location, dating method, mean annual precipitation, mean annual temperature, parent material and geomorphic surface.

Based on the reviewer's suggestion we have added additional language about the EEMT model at lines 184-187 in the revised manuscript.

The reviewer added comments about the influence of primary minerals on the sand fraction, variability of weathering, and poor prediction with regards to soil formation on amphibolite. The influence of parent material and lithology is discussed at length in lines 403-424 and lines 467-479 in the original manuscript. Further, we did not claim that the model was insensitive to parent material; we discussed the role of parent material and differential weathering rates as a possible explanation for poor model fits and suggest including parent material within the model apparatus a possible correction. The model makes no assumptions about the parent material, in that parent material is not directly expressed within the probability distributions, all soils are considered equally and global values are used to parameterize the probability distributions. We have added language to the revised manuscript at lines 417-419 about the dominance of primary minerals in the sand and silt fractions and have revised lines 580-581 to reflect the role of initial conditions within the model.

References Jenny, H. 1941. Factors of Soil Formation: A System of Quantitative Pedology. Dover Publications, Inc, New York, New York.

Jenny, H. 1961. Derivation of state factor equations of soils and

ecosystems. Soil Sci. Soc. Am. J.: 385–388Available at https://dl.sciencesocieties.org/publications/sssaj/abstracts/25/5/SS0250050385 (verified 29 January 2015).

Rasmussen, C., J.D. Pelletier, P.A. Troch, T.L. Swetnam, and J. Chorover. 2015. Quantifying Topographic and Vegetation Effects on the Transfer of Energy and Mass to the Critical Zone. Vadose Zo. J.Available at https://dl.sciencesocieties.org/publications/vzj/abstracts/0/0/vzj2014.07.0102.

Rasmussen, C., R.J. Southard, and W.R. Horwath. 2005. Modeling Energy Inputs to Predict Pedogenic Environments Using Regional Environmental Databases. Soil Sci. Soc. Am. J. 69(4): 1266–1274Available at https://www.soils.org/publications/sssaj/abstracts/69/4/1266 (verified 3 November 2014).

Rasmussen, C., and N.J. Tabor. 2007. Applying a Quantitative Pedogenic Energy Model across a Range of Environmental Gradients. Soil Sci. Soc. Am. J. 71(6): 1719Available at https://www.soils.org/publications/sssaj/abstracts/71/6/1719 (verified 27 October 2014).

Rasmussen, C., P.A. Troch, J. Chorover, P. Brooks, J. Pelletier, and T.E. Huxman. 2011. An open system framework for integrating critical zone structure and function. Biogeochemistry 102(1-3): 15–29Available at http://link.springer.com/10.1007/s10533-010-9476-8 (verified 21 October 2014).

---

## Author Response (AR1)

Response to Editor Peter Finke
Shepard C., Schaap MG., Pelletier JD., and Rasmussen C.

We thank the editor for his comments and recommendations on the manuscript titled "A probabilistic approach to quantifying soil property change through time integration of mass and energy input." We have responded to and addressed the editor's comments and remarks below and in the revised version of the manuscript.

**Response to general remarks:**
The model requires the inputs of time and effective energy and mass transfer (EEMT, Rasmussen et al., 2005; Rasmussen and Tabor, 2007; Rasmussen et al., 2011) to predict the probable range of a particular soil physical property at a given location. In this presentation, we focus on clay content as a physical property that reflects pedogenic change. EEMT is clearly rooted in the classical Jenny factorial approach for describing soil forming system, and as such does not describe any particular soil forming process. EEMT is a flux term, and is it not used here to parameterize a mechanistic model of pedogenesis as suggested by the editor. EEMT simply quantifies the energetic contributions from effective precipitation and net primary productivity as quantitative measures of climatic and biological forcing/input to the soil forming system. EEMT quantifies the energy transferred to the soil system that can perform pedogenic work, such as chemical weathering or carbon cycling, or any other soil forming process; it does not describe or quantify any one process, and this is indicated on lines 102-105.

Application of the present approach in the critical zone environment requires no soil information. Here we used an established geomorphically based numerical model that predicts local erosion rates, soil depth and local soil residence time from topography and a maximum rate of soil production (that can be assumed or based on local catchment derived denudation rates) (Pelletier and Rasmussen, 2009). The geomorphic model is mechanistic and process based, describing mass production and transport using established transport "laws". The editor expressed doubt of the ability to predict soil information in the critical zone environment; however, we present clear model results that this approach can be used to predict clay content completely independent of soil data. This is a key piece of soil information to understanding critical zone function and evolution. We further argued that the present approach can greatly inform our understanding of the distributions of soil physical properties and facilitate further hypothesis generation. For example, the present approach did not accurately predict clay stocks at specific locations within the Santa Catalina Mountains-Jemez River Basin granite sub-catchment (Lines 329-339); any number of hypotheses and questions can be formulated and tested as to why the model failed to predict clay stocks at these locations, and the current model formulation can be updated to accommodate these findings. Further, the present results suggest an incomplete understanding of the soil-landscapes within these catchments, which may not have been found by using techniques such as digital soil mapping.

The strength of the current approach lies in the ability of the model to capture all soil forming factors into one relatively simple mathematical apparatus. We make no claims of modeling particular soil forming processes, a fact that we state clearly in lines 145-149 and in lines 151-152. As true of any factorial treatment of soil systems, the model captures either the net effect of all considered soil forming processes or rather the implicit result of soil forming processes, by considering all soil profiles equally. This is the same foundation for any number of digital soil mapping exercises, as typified by the SCORPAN statement of McBratney and others – the model uses factors to predict soil properties and is not a mechanistic model of process. The model only indirectly captures soil forming processes by not restricting the model to any particular spatial or temporal extent or any particular parent material. We disagree with the editor's comment about the delineation of the model domain. By restricting the model domain, either spatially, temporally, or with regards to parent material, entire suites of soil forming processes would not be captured, limiting the applicability of the present approach.

**Response to Question 1. Why is the model forced to a bivariate pdf form? Techniques were on the shelf?**

A bivariate normal density distribution was used for the present approach because it generally represents the mathematically simplest bivariate distribution and is easily parameterized. However, we did not consider other bivariate density functions, we wanted to demonstrate proof of concept before exploring complexities or refinements to the approach. We have added language to the revised manuscript at lines 131, indicating that we choose to force the data to a bivariate normal structure and that other density functions are available and may provide a better fit to the soil physical property of interest.

**Response to Question 2. Question with regards to quantifying soil age and EEMT.**

The editor is correct in that the model assumes EEMT is constant over the duration of pedogenesis. Further, we agree with the reviewer that climate throughout the Quaternary has not been constant (Zachos et al., 2001), and that this inconsistency likely has influenced our predicted soil property values. We directly addressed this model limitation in section 4.3, lines 503-514:

"Furthermore, global climate patterns have shifted dramatically over the last 65 Mya (Zachos et al., 2001). The majority of soils observed in the compiled chronosequence database span the Quaternary, including both the Holocene and Pleistocene. The Pleistocene was marked by a number of major glacial-interglacial cycles at approximately 100,000-year intervals (Imbrie et al., 1992; Wallace and Hobbs, 2006), which corresponded with shifting climatic conditions, e.g., for large portions of the northern mid-latitudes glacial periods were generally cooler and wetter, and interglacial periods were warmer and drier (Connin et al., 1998; Petit et al., 1999). Further, the Pleistocene climate shifts likely influenced the rates of weathering and clay production (Hotchkiss et al., 2000). Taking into account the differences in past and modern climate would likely diminish disparities between observed and modeled soil physical properties. Reconstructed global paleo-EEMT values would improve model accuracy, and limit uncertainty in the probabilistic ranges of soil properties for soils older than Holocene age."

Further, we discussed a possible model correction in which paleo-EEMT values could be calculated and used to provide better estimates of TPE. However, in order to calculate an accurate accounting of paleo-EEMT values would require datasets of about past mean monthly air temperature, mean monthly precipitation and monthly net primary productivity (Rasmussen et al., 2005, 2011; Rasmussen and Tabor, 2007) for the entire duration of pedogenesis.

Unfortunately, few, if any, locations on the planet have spatially explicit paleoclimatic records with all the necessary data requirements to perform this calculation, although paleoclimate predictions are improving, e.g. the recent CIMP4 general circulation model application to predict global LGM climates, which represents an ongoing opportunity to incorporate such data into pedogenic models. As such, we made the simplifying assumption that the current climate can be used to represent of climates that many of the included soils evolved under. This is true of any factorial approach and representation of soil data that includes soils older than the Holocene. Any representation of soil properties relative to mean annual temperature or precipitation or any plot of soil property change vs time invariably includes past climate variation influence of soil property evolution. We clearly stated and recognized this in the text.

Defining soil age is a challenge in many landscapes as the editor suggests; however, there are simple techniques that can be used to estimate soil age without the need for expensive cosmogenic radionuclide dating. The age of geomorphic landforms can be estimated by using the cross-sectional shape of gully cuts or scarp-like surfaces and hillslopes and a known hillslope diffusivity value (Bucknam and Anderson, 1979; Hsu and Pelletier, 2004; Pelletier et al., 2006; Pelletier and Cline, 2007). Estimating geomorphic landform age requires only the use of either a digital elevation model (DEM), or profiles of scarp elevation, both of which are easily and inexpensively attained. Further, relative age dating is widely used in chronosequence studies and provides general estimates and constraints of soil-geomorphic surface ages (Schaetzl and Anderson, 2005). With regards to upland catchments, catchment averaged denudation rates can be estimated from cosmogenic radionuclides (CRN) using a smaller number of samples than would be necessary for quantifying full CRN depth profiles (Granger et al., 1996). Using a geomorphically based model of soil depth, spatially explicit soil ages can be calculated was discussed in lines 220-224. As such, we do not agree with the editor the target variables are more easily determined than soil age; for any chronosequence study, soil age would have to be determined regardless of the target variables of interest, assuming soil age is too expensive or indeterminable is not an appropriate or accurate critique of the presented model.

**Response to Question 3. Removal of buried horizons.**

Buried horizons were removed from the dataset, as we assumed that these buried horizons were not reflective of the relationship between the modern climate and the subaerial soil horizons. We decided to remove buried horizons, as the subaerial soil horizons likely are more correlated with the current climate, as compared to the buried horizons. Eolian horizons were removed from soils described in McFadden et al. (1986), because these horizons are likely significantly younger than the basaltic flow dates that were used to represent the soil age; however, based on the reviewer's suggestion we have removed these soils from the current dataset and updated lines 194 to reflect this change. Only buried horizon have been removed from the dataset presented in the revised manuscript.

**Response to Question 4. Non-linear or transient soil formation.**

We specifically addressed non-linear soil formation and internal or intrinsic feedbacks driving soil development in section 4.3, lines 490-502:

"Internal or intrinsic feedbacks and thresholds within the soil system drive pedogenic development without changes in the external state factors (Muhs, 1984; Chadwick and Chorover, 2001). For example, greater chemical weathering and clay production due to increased water residence time caused by argillic horizon development is the result of an internal feedback that is independent of the external climatic and biological system (Schaetzl and Anderson, 2005). These thresholds can operate as progressive or regressive processes, driving soil formation forward or hindering further development (Johnson and Watson-Stegner, 1987; Phillips, 1993). Internal soil development feedbacks were not explicitly considered in the present model formulation. The presence of these internal feedbacks may partially explain error within the model predictions. Changes in EEMT would not explain all observed differences in soil properties over the age of the soil. However, if these feedbacks were operating in the included soils, the influence of intrinsic thresholds was implicitly captured within the probability distributions, partially accounting for the role of internal soil development feedbacks on soil formation."

The model does capture all soil forming processes implicitly, in that no one process is explicit expressed or quantified. Further, we agree with the reviewer that the model does produce a prediction of soil physical properties based on the net effect of these soil forming processes. We have edited lines 149-153 by removing the word "implicitly".

**Response to Question 5. Human impacts on soil formation.**

We did not address human impacts within the current manuscript or even discuss anthropogenically driven changes in land use or climate. Here we demonstrate the use of a probabilistic model for quantifying the distribution of soil properties we observe currently on the Earth's surface that has arisen during the Quaternary. Furthermore, the energetic contributions from human impacts and dust influx can and have been incorporated within the EEMT apparatus (Rasmussen et al., 2011). The energetic inputs from dust or fertilizer additions, for example, are generally orders of magnitude smaller than the energetic inputs from solar radiation, precipitation, or primary productivity into the soil system. The energetic inputs to the soil from other direct human activities such as the compression of soil due to farming equipment or the increased erosion due to construction or plowing (Rasmussen et al., 2011). We have added language to indicate the adaptability of the EEMT model for differing soil environments at lines 212-215 in the revised manuscript. In specific systems, both dust inputs and human impacts may be significant, however, the vast majority of the soils included in the presented dataset are not directly impacted by human activities or modern dust influx. As the model is probabilistic in nature, the model can simply predict a probable range of target soil physical properties, the domain is generally unconfined. As stated above, human and dust inputs to the soil system can be incorporated into EEMT allowing the inclusion of these soils within the model. Furthermore, the application of the model in this manuscript was to predict clay content, this is a soil property that does not readily change over human time-scales – but rather reflects geologic time scale pedogenic change.

**Response to Question 6. Regards to use of energetic models.**

Energetic approaches to quantifying soil physical properties and soil formation are able to deal with differing mixtures of the soil forming factors. The soil state factor model has the potential to be expanded beyond the classical five soil forming factors to include influences from site-specific soil forming factors such as the addition of fertilizer to soils or increased erosion due to human activity (Jenny, 1941, 1961). All energetic pedogenic models are derived from the soil state factor model (Minasny et al., 2008), and as such have the potential to be expanded to accommodate additional soil forming factors. Energetic approaches account for the potential fluxes of matter and energy into the soil system that are associated with the soil forming factors, relating the energetics of the fluxes into the soil system to soil physical properties and structures (Volobuyev, 1964; Runge, 1973; Smeck et al., 1983; Rasmussen et al., 2005, 2011; Rasmussen and Tabor, 2007). The energetic input from fertilizer can be easily quantified and included with the model scheme for EEMT when appropriate, however, the energetic additions to the soil from fertilizer are orders of magnitude smaller when compared to the energetic additions of soil radiation, effective precipitation, and net primary productivity (Rasmussen et al., 2011).

The impact of anthropogenic climate change on soil physical properties can be incorporated into the EEMT model space. EEMT can be updated to include the impacts of increased atmospheric $CO_2$ on the fluxes of matter and to the soil system. Local changes in air temperature, precipitation, evapotranspiration, and net primary productivity due to increased atmospheric $CO_2$ are quantifiable, and can be easily incorporated into the EEMT model. Further, EEMT can be calculated on a range of temporal scales from near-real time through the use of eddy flux tower and meteorological flux measurements to annually (Rasmussen et al., 2015). With EEMT values updated to include the impacts of anthropogenic climate change, the presented model structure is capable of incorporating these influences on soil formation. As such, we disagree with the reviewer that energetic-based pedogenic models are not capable of handling changing earth surface conditions, and can be updated to accommodate human influences on the climate and landscapes.

**Response to specific remarks**

**Response to remarks on Lines 94-97, interchangeability of $P_x$ and $L_o$:**

The soil state factor model developed by Jenny (S=f(cl,o,r,p,t)) was later formulated by Jenny as S=f($P_x$,$L_o$,t), recognizing that climate and biology are generally flux inputs into the soil system and relief and parent material are site factors. $P_x$ influences pedogenesis and soil evolution over the lifetime of the soil and may be time dependent, whereas $L_o$ generally represents the initial state of the soil forming system and is not time dependent (Jenny, 1961). Interchanging the influence of $P_x$ and $L_o$ is not possible. Relief or topography can vary over time, and in certain formulations of the soil state factor model may be considered time dependent, however, the chronosequence data used to parameterize the present approach are sited on low sloping surfaces, and changes in topography were minimized. Furthermore, as described by Jenny, and approximately 70 decades of soil science, the soil state factors do not describe soil forming processes, the state factors only describe the soil forming environment, i.e. climate is not a soil forming process, but a description of the conditions under which certain soil properties are observed and soil processes operate.

**Response to remarks on Lines 182-183: Depth weighted percent clay calculation and bulk density values**

We used depth weighted average percent clay in the prediction of clay stocks to account for the greater influence of thicker soil horizons on the account of clay stocks. By calculating a depth weighted average we are accounting for the distribution clay with depth, and summarizing those values into one value. Our model was trained on depth weighted average clay percentages from the chronosequence database; consequently, we also used depth weighted average clay values for predicting clay on a mass per area basis. We agree with the editor, bulk density is not constant throughout a profile, unfortunately, bulk density is difficult to measure, or is often not measured in the field. Further, bulk density data are not commonly reported within the soil science literature or in the available chronosequence literature. Without the necessary data, we chose to assume a constant value of 1500 kg m$^{-3}$ for all soil profiles used in the calculation of predicted mass per area clay. If bulk density data were available, those data could be easily included in the prediction of mass per area, and likely the presented probabilistic-energetic would likely better predict clay stocks.

**Response to remarks on Lines 238-240: Assumption of 1500 kg m$^{-3}$ bulk density and use of RF% for calculating clay stocks.**

Bulk density is not a commonly measured soil variable as it is often difficult to obtain measurements for bulk density from soil profiles, and values for bulk density are highly method dependent; there was low reporting of bulk density estimate in the available chronosequence literature. Due to a lack of measured values, a constant value was chosen for all profiles; if bulk density measurements are available than the measured values should be used in the predictions of clay stocks. Further, RF% data were used in the predicted clay stocks, as (1-RF%) in Eq. 9 describes the volume or fraction of the soil profile in which clay sized particles accumulate. Additionally, RF% did not influence the prediction of clay stocks (line 326); if RF% data were unavailable, a standard or constant value could be assumed for predicting clay stocks. Further, these simplifications for calculating soil properties on a mass per area basis are standard corrections and assumptions that are made throughout the available literature. With missing or incomplete data, the complexities of measuring soil properties in the field, educated assumptions are usually required.

**Response to remarks on Section 3.1 Bias in sampling and stratification of L$_o$:**

The editor did not fully understand the model background as presented in the manuscript. L$_o$ was not used for stratification; L$_o$ is not directly expressed within the model structure. We removed L$_o$ from Eq. 4 to produce Eq. 5, justifying this simplification as time partially accounts for the influence of topography and parent material variation. Soil residence time on a landscape is proportional to slope or curvature (Heimsath et al., 1997, 2002; Yoo et al., 2007). Additionally, the degree of weathering or alteration of the parent material and the presence of secondary minerals and products are also proportional to the soil residence time (Brimhall and Dietrich, 1987; Chadwick et al., 1990; Brimhall et al., 1992). We chose to break down model predictions using leave one out cross validation by parent material in Fig 4 to demonstrate the model was insensitive to different parent materials or landforms, the predictive ability of the model did not vary significantly between the 3 broad parent material categories. We did not calculate model parameters based on parent material, we presented global parameter values in Table 2. We have updated lines 124 and lines 155-157 to clarify this point.

Biased sampling due the use of chronosequence studies is an issue faced in all of soil science. Soil pits are generally preferentially sited in locations where it is possible to dig a soil pit of sufficient depth to sample the soil profile. Any chronosequence study or synthesis of chronosequence data is hampered by biases within soil sampling and presentation of selected data in the literature. Biases in estimated model parameters are based upon sampling techniques and availability of chronosequence data in the literature not due to selective sampling of chronosequence data used to calculate model parameters. We did not limit the data used to calculate the model parameters from the chronosequence literature as a way to minimize errors within the presented model.

**Response to remarks on Line 346, clay content change:**

In lines 151-152 we stated: "The model assumes all changes in soil physical properties are due to pedogenic processes.", and in lines 345-348 we stated: "Weathering and clay production are primary pedogenic processes (Birkeland, 1999; Schaetzl and Anderson, 2005), and because the model assumed all changes in the soil profile are due to these processes, the model was the most effective at predicting clay content." These statements are in agreement with each other. The model implicitly captures the net effect of all pedogenic processes, we assumed that all changes in the soil profiles are due to pedogenic processes, and the primary pedogenic processes are weathering and clay production, amongst a suite of other processes (Schaetzl and Anderson, 2005), meaning the model captures the influence of weathering and clay production in soil property change. There is no "pretending", we are not misrepresenting the model or its predictive power in any way. We did not claim to model soil forming process, only the end result of all soil forming processes at specific times and energetic inputs based on the available chronosequence literature; this is true for any chronosequence or time based representation of soil data that has been published over the last 70+ years.

**Response to remarks on Section 4.1.1, and where the model underperformed:**

In many areas, estimated TPE values likely do not account for the total flux of mass and energy into the soil system. Error in predicted clay percentages are likely partially a function of error in TPE estimates. We discussed underestimations of TPE due to changing climate, topography, parent material differences and intrinsic thresholds in soil formation in Section 4.3. Further, we did not constrain the chronosequence data set used to calculate the model parameters, as the formulated model is capable of handling soil data from a wide range of environments and locations due to its probabilistic component. We highlighted where the model failed to predict soil property values as a way to highlight locations where we still have an incomplete understanding of soil formation, or places where parent material greatly influences resultant soil formation (i.e. coral reef terraces). The inability of the model to predict soil properties in particular areas, suggests that a soil-forming factor not included in EEMT or TPE is highly influencing soil formation in this area, not that input data used to calculate model parameters need to be constrained. Models are only representations of reality and there is no logical need for a model to perform perfectly. It is generally beneficial to identify locations and conditions under which models do not work, as a way to identify potential model refinements.

We did not highlight model failures as an excuse to selectively choose data to achieve the best model predictions as suggested by the editor. Further, constraining data to achieve a successful model prediction is uninteresting, as one cannot identify locations or conditions under which the model breaks. Without the inclusion of coral reef terrace chronosequence, we would not have identified that the model has an inability to predict resultant soil formation under fine textured parent materials and tropical climates.

**Response to remarks on Line 439, model assumptions about initial conditions:**

The editor did not understand the model background as written in lines 115-127, we did not stratify the data by parent material:

"Explicitly including time in Eq. 4 through TPE partially captures variation in soil property change attributable to topography and parent material. Soil residence time may be directly related to landscape position through topographic control on soil production and sediment transport/deposition (Heimsath et al., 1997, 2002; Yoo et al., 2007). Additionally, parent material modulates soil residence time through control on soil depth (Rasmussen et al., 2005; Heckman and Rasmussen, 2011), soil production, and sediment transport rates (Andre and Anderson, 1961; Portenga and Bierman, 2011). The initial conditions of the soil forming system ($L_o$) are never fully known; however, representing the state of the soil system as a probable distribution of values, implicitly accounting for soil age, and not constraining the initial soil forming conditions, Eq. 4 can be stated simply as:

$$\mathbb{P}(s_1 \leq S \leq s_2) = f(TPE) \qquad (5)$$

where the probability state of the soil, $\mathbb{P}(s_1 \leq S \leq s_2)$, bounded by a lower and upper limit, is a function of one quantifiable variable."

We simply removed $L_o$ from Eq. 4 to write Eq. 5, TPE partially accounts for variation in $L_o$ due to the influence of topography and parent material on soil residence time, as discussed above. We did not use parent material to stratify model parameter estimates; we calculated model parameters for the entire chronosequence dataset. We did not make assumptions about the soil parent material, or include any data about the parent material within the presented model. The statement in Line 439 is accurate. We have updated lines 124 and lines 155-157 to clarify this point.

**Response to remarks on Line 459, potential to model landscape evolution:**

We strongly disagree with this statement. First, the application of the model that couples the geomorphic model of Pelletier and Rasmussen (2009) explicitly includes soil production and sediment transport to predict landscape variation in soil depth and residence time – these values coupled with TPE yield estimates of soil physical properties completely independent of any soil data. This coupled model may be used to predict soil and landscape evolution across any range of topographic and/or climate scenarios, and yield probabilistic estimates of soil clay content. As stated throughout the manuscript, the model was designed to capture Quaternary soil evolution; additionally, the focus on clay content necessitates a geologic time scale perspective as this property changes not on human scales, but on pedogenic time scales. Changes induced by human activity could be incorporated into the sediment transport of the Pelletier and Rasmussen (2009) model, as well as incorporated into the energy and mas transfer terms, but in terms of changes in clay content over human time scales these changes will likely be insignificant. As stated in the manuscript, this approach could be used to investigate potential landscape scenarios. Using the geomorphic model (Pelletier and Rasmussen, 2009), potential landscape evolution scenarios could be investigated where changes in topography and soil thickness are used to determine changes in soil properties across small watersheds. We stress that potential landscape evolution scenarios could be investigated, assuming the landscape is at steady state, soil development and evolution could be teased apart using the presented approach; any predictions drawn from such hypothetical modeling exercises would only be a potential future for any landscape. Furthermore, as discussed in this review EEMT can be updated to include the influence from human impacts on the atmosphere and landscapes, and TPE can also be calculated to include the influence of a changing climate (Rasmussen et al., 2011).

As stated in the manuscript and repeatedly above, we used modern EEMT integrated over the age of the soil as the estimate of "total" pedogenic energy input as the best available data that we have. We clearly recognize that this does not incorporate past climate change, leading to what could be under/over estimates of TPE depending on how the local climate system changed at each included location during glacial periods. The majority of sites were from northern hemisphere mid-latitude sites suggesting modern EEMT, and hence, TPE likely underestimates the total pedogenic energy transferred to each location. As noted spatially explicit estimates of LGM climate conditions are now available, but we currently lack a time resolved estimate of paleoclimate variation. As such, we used modern values as proxies for soil forming factors. This is true for any study of soil properties relative to modern climate. Based on the editor suggestion we have removed the reference to investigation of potential soil forming environments at lines 540-541 in the revised manuscript.

Lines 128-131; 152-154. We choose the bivariate normal density function for its simplicity and ease of parameterization. We did not consider other bivariate distributions. We wanted to demonstrate the use of bivariate probability functions for modeling soil properties from a probabilistic viewpoint. We have added language to lines 134-136 to indicate that only the bivariate normal density function was considered for the present approach.

Lines 144-154: Modeling soil properties over time requires a number of assumptions, as such every soil formation model is an approximation of reality. We made these assumptions to reduce model complexity and to make the model as mathematically simplistic and easily parameterized as possible. We are aware of many issues with these assumptions, and we discuss at length the implications of these assumptions in lines 464-514 on model outputs and model failures. We disagree with the reviewer that these assumptions are unrealistic, as the present approach is effective at prediction soil property across wide variety of environments and ecosystems.

Line 159: We have replaced "over" with "more than" based on the reviewer's comment.

Line 161-162: We have edited the manuscript to reflect the reviewer's comment, and removed one of the phrases "within the present study" from the revised manuscript.

Lines 164-166: We agree with the reviewer that Southern Hemisphere and mid-latitude sites are underrepresented within the current dataset; however, we are limited about the availability of published datasets. A number of studies from South America, Africa and the Tropics were initially identified, but only a small number of these studies included horizon-level texture data or numerical or approximate ages for the described soil profiles.

Lines 168-177: Based on the reviewer's suggestion we have added additional explanation and description of EEMT, at lines 184-187 in the revised manuscript.

Lines 182: We agree there are many chemical and biological changes that occur over time that are not dependent on soil textural changes, we do discuss intrinsic changes in soil properties from lines 490-502. We have updated the manuscript based on the reviewer's comments, replacing the word "proxies" with "examples of soil property change with time".

Line 213: We have deleted the first "terrain" based on the reviewer's comment at lines 236 in the revised manuscript.

Lines 232-236, Lines 389-390: As discussed in the manuscript and acknowledged by the reviewer, soil depth is correlated to and dependent upon topography and hillslope redistributive processes. Soil depth varies systemically across hillslope as indicated by the contemporary work of Dietrich, Heimsath, Pelletier, amongst many more, and was discussed in the manuscript. We only stated that soil depth incorporates the strength of these processes in lines 232-234, and we have changed "correct" to "incorporate" based on the reviewer's comments. Further in lines 389-390, we simply state the soil depth acts as a proxy for hillslope processes, not that soil depth accounts for all hillslope processes or the complexity of sediment redistribution on a hillslope. We acknowledge the incomplete understanding of soil depth and weaknesses of soil depth predictions at lines 335-336. Soil depth only partly accounts for the complexity of hillslope processes.

Lines 248-263: The outputs from the process-based numerical soil depth model and the topographically resolved EEMT model were used to calculate the necessary model inputs to the probabilistic model. Soil depth was used to calculate soil residence time, and TPE values were calculated from topographically resolved EEMT and soil residence time values. TPE values were used in the probabilistic model to calculate depth weighted clay content values, and Eq. 9 was used along with predicted soil depth values to calculate mass per area clay across the small-forested catchment. Based on the reviewer's suggestion we have added language to clarify Section 2.4.1 from lines 306-309 in the revised manuscript and a flow chart to the revised manuscript.

Line 271: We updated line 271 to indicate that Pearson's correlation was used to parameterize the probability distributions. Further, we updated lines 276 and 281 to indicate the Pearson's correlation is represented in the reported statistics.

Lines 322: We have added the word "regression" to indicate that we are discussing the slope of the regression line presented in Fig 6b.

Lines 359-372: We agree with the reviewer's comment, the sand and silt fractions are both dominated by resistant primary minerals. We have added a statement to the revised manuscript at lines 423-425.

Lines 377-379: We have added additional references that indicate the non-linear dynamics of soil depth and soil deepening at lines 407 in the revised manuscript.

Lines 413: We agree with the reviewer that scale is likely an important factor in predicting soil properties. Finer spatial scales will likely better match the local variation in soil properties, but may also lead to greater potential for prediction errors. Further, finer scale information about local lithology differences and weathering rates are likely required. We have added discussion of the issue of scale in predicting soil properties in lines 464-467 in the revised manuscript. Further, the issue of scale in lithology and weathering rates is discussed from lines 468-479 in the revised manuscript.

Lines 421-424: We have added language clarifying the difficulty of including differing weathering rates based on lithology to the revised manuscript at lines 477-478 in the revised manuscript.

Lines 426-462: The model predicts ranges of clay contents. The bivariate normal distribution predicts the conditional parameters of a univariate normal distribution for the soil property of interest. With the conditional univariate parameters, the model user can determine the probability of observing a particular range of clay values. This approach represents a first attempt at a true probabilistic prediction of soil property values, more complex probabilistic approaches that incorporate explicit change with time are possible. However, these more complex probabilistic approaches would require an equation over which probabilistic predictions can be updated over time.

Lines 464-514: With the appropriate updates and additions to the probabilistic model many of these caveats and issues with the model are correctable. Further, many of these caveats specifically address the assumptions and simplifications that are discussed in lines 174-185 in the revised manuscript.

Lines 467-470: We agree with the reviewer's comment, parent material can greatly influence rates of pedogenesis or weathering, regardless of controlling for the other soil forming factors. We have added language to the revised manuscript at line 527 to reflect this issue.

Table 2: We have updated the caption for Table 2 with the explanations of the column headings. We use a "rho" or $\rho$ to represent Pearson's correlation.

Response to Anonymous Reviewer #2

We thank the reviewer for their comments on the manuscript titled: "A probabilistic approach to quantifying soil property change through time integration of energy and mass input". Below we have detailed our response to the reviewer's comments, including how the manuscript was edited.

We disagree with a number of the assertions made by the reviewer. The objective of the presented work was to present the development of a probabilistic model of soil property change using time integrated energy and mass inputs. We added language clarifying this hypothesis at lines 78-80 in the revised manuscript. We think the title is appropriate for the manuscript as written; the manuscript discusses modeling the change in soil properties using energy and mass inputs integrated with time or the age of the soil system, as such we did not change the manuscript title. Further, we did not attempt to model any soil forming process. The presented probabilistic approach is based upon the soil state factor model, which only considers the state of the soil forming environment to understand soil formation or soil structures (Jenny, 1941, 1961). Similarly, the effective energy and mass transfer model only quantifies the amount of potential heat and chemical energy added to the soil, it does not describe any soil forming processes (Rasmussen et al., 2005, 2011, 2015; Rasmussen and Tabor, 2007). This is clearly stated in lines 85-87 and 145-152 in the original manuscript. The model simply quantifies the net effect of all soil forming processes operating within the soil forming system. The revised manuscript has been edited to clarify this point at lines 154. Further, the discussion focused on the reasons for poor or acceptable model fits for the modeled soil properties, and discusses advantages and disadvantages to the probabilistic approach. Nowhere within the discussion do we claim that the presented approach models any soil forming process or soil formation. As was stated in lines 145-152 of the original manuscript, the model assumes all changes within the soil are due to pedogenic processes, this aligns with what is stated in the discussion in lines 345-348 in the original manuscript. EEMT and TPE quantify the amount of energy added to the soil system capable of doing pedogenic work, e.g. chemical weathering, which is most representative of clay formation.

The influence of climatic variability on the model results is discussed in length in lines 503-514 in the original manuscript. While palecolimatic reconstructions are available, such as the spatially explicit LGM paleoclimate reconstruction from the CIMP4 general circulation model, time resolved paleoclimatic reconstructions are still largely unavailable for many locations. We chose to use modern climate values as they represent the best available data. We did not discuss human or anthropogenic impacts on soil formation or evolution. We are exclusively focused on Quaternary soil formation. In this manuscript, we focused on clay formation, which occurs on a geologic timescale and does not readily change on human timescales. Additionally, the EEMT model is easily adaptable to accommodate human influences on the energy and mass inputs into soil, such as fertilizer inputs (Rasmussen et al., 2011).

We have added the age span and depth ranges to the revised manuscript at lines 179-180. Data on vegetation and stoniness were not available across all the sequences and were not included. Further, many of these data are included in Table S1, where all of the chronosequences are listed along with location, dating method, mean annual precipitation, mean annual temperature, parent material and geomorphic surface.

Based on the reviewer's suggestion we have added additional language about the EEMT model at lines 184-187 in the revised manuscript.

The reviewer added comments about the influence of primary minerals on the sand fraction, variability of weathering, and poor prediction with regards to soil formation on amphibolite. The influence of parent material and lithology is discussed at length in lines 403-424 and lines 467-479 in the original manuscript. Further, we did not claim that the model was insensitive to parent material; we discussed the role of parent material and differential weathering rates as a possible explanation for poor model fits and suggest including parent material within the model apparatus a possible correction. The model makes no assumptions about the parent material, in that parent material is not directly expressed within the probability distributions, all soils are considered equally and global values are used to parameterize the probability distributions. We have added language to the revised manuscript at lines 417-419 about the dominance of primary minerals in the sand and silt fractions and have revised lines 580-581 to reflect the role of initial conditions within the model.

Table 2: We have updated the caption for Table 2 with the explanations of the column headings. We use a "rho" or $\rho$ to represent Pearson's correlation.

**A probabilistic approach to quantifying soil physical properties via time-integrated energy and mass input**

[revised manuscript text omitted]

Applying this approach required certain assumptions and simplifications. The model assumes that climate was constant over the entire duration of pedogenesis. The model makes no assumptions about the progressive and regressive processes that drive pedogenesis; by weighing all profiles equally, the net effects of both progressive (e.g., horizonation, clay accumulation, reddening, etc.) and regressive (e.g., haplodization, erosion, pedoturbation, etc.) pedogenic processes (Johnson and Watson-Stegner, 1987; Phillips, 1993a), are captured in the model structure. The model also does not consider the net effect of progressive and regressive pedogenic processes on the distribution of selected soil properties with depth. The model makes no assumptions about the initial soil forming system, and we did not constrain the model to any particular initial condition for either parent material or geomorphic landform; the model simply describes the probability of a location exhibiting a range of soil properties based on TPE. The model assumes all changes in soil physical properties are due to pedogenic processes. We used a bivariate normal distribution; consequently the model assumes the data conforms to a normal distribution.

**2. Methods**

**2.1 Data collection and preparation**

The probability distributions were parameterized using an extensive literature review of chronosequence studies. More than 140 chronosequence publications were identified using

Christopher Shepard 2/7/17 1:39 PM

Christopher Shepard 2/2/17 12:10 PM
Christopher Shepard 11/17/16 12:05 PM

Christopher Shepard 11/17/16 12:05 PM

Christopher Shepard 12/19/16 9:56 PM

Google    Scholar    (scholar.google.com)    and    ThomsonReuters    Web    of    Science (webofknowledge.com), forty-four of which contained the required data. Inclusion within the present study required: profile descriptions with horizon-level clay, sand, and silt content, soil depth; well-defined ages of the soil-geomorphic surfaces; and geographic coordinates or maps showing locations of the described profiles. The chronosequences spanned a wide range of geographic locations, ecosystems, climates, rock types, and geomorphic landforms (Fig 1, Table

S1). The chronosequence soils spanned ages from 10 years to 4.35 Myr and depth ranges from

3.0 cm to 1460 cm, with mean annual temperature and precipitation ranging from -11.2 to 28.0

°C and 3.0 to 400 cm yr$^{-1}$, respectively. We were limited in site selection by the available data; as such we could not control for any bias that may exist with regards to site selection and reported soil property values.

**2.2 Total Pedogenic Energy**

The influence of both climate and vegetation at the locations of each soil profile was determined using effective energy and mass transfer (EEMT) (Rasmussen and Tabor, 2007;

Rasmussen et al., 2005). EEMT quantifies the heat and chemical energy from effective precipitation and net primary productivity added to the soil system (Rasmussen and Tabor, 2007;

Rasmussen et al., 2005, 2011). EEMT describes the energy added to the soil system that can perform pedogenic work, such as chemical weathering and carbon cycling. EEMT is adaptable to include specific energetic inputs to the soil system based upon the prevailing soil forming environment, e.g. the energetics from added fertilizer in an agriculture field or the impact of human induced erosion (Rasmussen et al., 2011). The EEMT values for each soil profile were extracted from a global map of EEMT derived from the monthly global climate dataset of New et

Christopher Shepard 2/15/17 5:12 PM
Christopher Shepard 12/19/16 10:09 PM

al. (1999) at 0.5°x0.5° resolution using ArcMap 10.1 (ESRI, Redlands, CA) (Rasmussen et al.,

2011). For the chronosequence soils, EEMT values ranged from 2,235 to >200,000 kJ $m^{-2}$ $yr^{-1}$.

Total pedogenic energy (TPE, J $m^{-2}$) was derived simply by multiplying EEMT (J $m^{-2}$ $yr^{-1}$) for each soil profile by its reported age (yr). TPE was used because it was a better predictor of soil physical properties relative to mean annual temperature, mean annual precipitation, or net primary productivity (Table 3).

**2.3 Application to chronosequence data**

The chronosequence database included 44 distinct chronosequences representing 405

[revised manuscript text omitted]

2d). We used calculated TPE from the topographically-resolved EEMT and soil residence time values to predict DWT clay, and coupled predicted DWT clay values with modeled depth from

Pelletier and Rasmussen (2009a) in Eq. 8 to predict mass per area clay at 2 m pixel resolution; the data processing and model apparatus are shown in Fig 3. We assumed a constant 50% rock fragment value for each location. The coupled geomorphic-TPE model outputs were compared with point measures of mass per area clay from Holleran et al. (2015) and Lybrand and

Christopher Shepard 2/7/17 1:39 PM

Christopher Shepard 12/22/16 10:49 AM

Christopher Shepard 12/22/16 10:49 AM

Christopher Shepard 2/15/17 5:19 PM

Christopher Shepard 2/7/17 1:40 PM

Rasmussen (2015). Model data were completely independent from the Holleran et al. and

Lybrand and Rasmussen datasets such that they served as validation data for the modeled output.

**2.5 Model domain**

The model was parameterized using chronosequence studies; as such, the model is best suited for generally low, sloping terrain. The model was extended to complex terrain using the described correction above (Section 2.4), widening the model domain to steeply sloping terrain.

The model does not consider human activities or aeolian additions, and should not be extended to soils significantly impacted by either humans or dust. The model was trained on a diverse array of parent materials and ecosystems, and could be utilized in climates with MAT ranging from -

10 to 28°C and MAP ranging from 3 to 400 cm $yr^{-1}$. The model could be utilized on soils spanning multiple magnitudes in age, from 10 yr to greater than 4Myr.

**3. Results**

**3.1 Application and parameterization to chronosequences**

[revised manuscript text omitted]

Christopher Shepard 2/7/17 1:40 PM

Christopher Shepard 12/22/16 3:44 PM
Christopher Shepard 12/22/16 3:44 PM
Christopher Shepard 12/22/16 3:44 PM
Christopher Shepard 2/8/17 11:37 AM
Christopher Shepard 12/22/16 3:44 PM
Christopher Shepard 2/8/17 11:38 AM

Christopher Shepard 12/22/16 3:44 PM
Christopher Shepard 2/8/17 11:38 AM
Christopher Shepard 12/22/16 3:44 PM
Christopher Shepard 2/8/17 11:38 AM
Christopher Shepard 12/22/16 3:44 PM
Christopher Shepard 2/8/17 11:38 AM
Christopher Shepard 2/8/17 11:38 AM
Christopher Shepard 12/22/16 3:44 PM
Christopher Shepard 2/8/17 11:39 AM
Christopher Shepard 12/22/16 3:45 PM
Christopher Shepard 2/8/17 12:39 PM
Christopher Shepard 2/8/17 12:39 PM

[revised manuscript text omitted]

However, by combining probabilistic approaches parameterized using known landscapes, and geomorphically based landscape evolution models, predictions of the current state of the soil- landscape can be investigated. As was demonstrated in Fig 7B, combining the present approach with geomorphically based soil depth models generated from DEMs has great potential to predict soil properties across a diverse range of environments, without needing prior knowledge of the landscape other than topography and climate. Further, potential soil-landscapes can be investigated by updating EEMT values to incorporate future climate scenarios available from predictive climate models (Gent et al., 2011; Taylor et al., 2012) and topographic and hydrological impacts due to changes in topography over time (Rasmussen et al., 2015).

**4.3 Limitations and potential refinements**

There are obvious limitations within the current model: lack of consideration of parent material influences, topographic variation, human impacts, internal soil feedbacks and thresholds, determination of landscape and soil age, and differences in paleoclimate variation.

Parent material control on the relative proportion of weatherable minerals and mineral

Christopher Shepard 12/27/16 8:41 PM

Christopher Shepard 2/15/17 4:02 PM

weathering rates (Jackson et al., 1948) can manifest as vastly different soil morphologies and rates of pedogenesis when controlling for other soil forming factors or even without controlling for other factors (Heckman and Rasmussen, 2011; Parsons and Herriman, 1975; Phillips, 1993b).

The current approach implicitly assumes no information about the initial conditions, only that all clay production is a pedogenic process. Applying this approach to parent materials, where a large fraction of clay-sized particles formed through non-pedogenic processes, is thus limited and may explain why the model was ineffective for some soils. Refining the current approach would require normalization of soil to the particle size distribution of the soil parent material. Past studies have utilized highly characterized parent material data to model soil property change with time (Chadwick et al., 1990; Harden, 1982), but these data are generally difficult to obtain and often not reported in the available chronosequence literature.

Topography dictates soil chemical and physical properties and residence times, especially in complex terrain (Almond et al., 2007; Egli et al., 2008; Lybrand and Rasmussen, 2015), where non-linear diffusive hillslope processes control the fluxes of matter and energy into and out of the soil system (Heimsath et al., 1997; Pelletier and Rasmussen, 2009a; Rasmussen et al., 2015;

Yoo and Mudd, 2008b; Yoo et al., 2007). Using earlier versions of EEMT (Rasmussen and

Tabor, 2007; Rasmussen et al., 2005), the current formulation of the model and TPE does not explicitly quantify topographic variation, which may account for error within current soil property distributions and predictions. With the inclusion of topographic variation in EEMT

(Rasmussen et al., 2015) and topographic control of soil residence times (Foster et al., 2015;

West et al., 2013), we were able to correct this error with the present approach and effectively predicted clay stocks in complex terrain.

Human activities significantly alter soil physical properties (Grieve, 2001; Neff et al.,

2005; Pouyat et al., 2007). For example, differences in land use and increased grazing activity can alter soil physical properties such as clay and sand content across landscapes (Neff et al.,

2005; Pouyat et al., 2007), or compaction from farming equipment leading to increased bulk density and increased erosion rates (Fullen, 1985; Hamza and Anderson, 2005). Human impacts on soil physical properties were not included in the presented model. The energetic contributions due to human impacts can be incorporated within the EEMT apparatus, and adjusted model parameters can be calculated (Rasmussen et al., 2011). Human impacts on soil physical properties may be locally important, but for the majority of locations, human energetic contributions to the soil system are generally orders of magnitude smaller compared to the energetic inputs from solar radiation, precipitation, or primary productivity.

[revised manuscript text omitted]

**Acknowledgements**

We thank Molly Holleran, Rebecca Lybrand, and Ashlee Dere for providing data for this study.

Support for C.S. was provided by the University Fellows program at the University of Arizona and by the University of Arizona/NASA Space Grant Graduate Fellowship. This research was funded by the U.S. National Science Foundation grant no. EAR-1331408 provided in support of the Catalina-Jemez Critical Zone Observatory. LiDAR data acquisition was supported by U.S.

National Science Foundation grant no. EAR-0922307 (P.I. Qinghua Guo).

[revised manuscript text omitted]

---

## Editor Decision (ED1)

Manuscript "A probabilistic approach to quantifying soil property change through time integration of energy and mass input", by Christopher Shepard, Marcel G. Schaap, Jon D. Pelletier, and Craig Rasmussen

Decision on the manuscript after 3 reviews and 3 responses.

I thank the authors for their extensive responses to the reviewer's comments. A number of changes to the manuscript were already announced, and I am looking forward to read the revised text and evaluate these changes.

Additional, I am of the opinion that a number of comments were not yet fully or adequately addressed and ask the authors to respond to these in the manuscript as well.

**Re: Anonymous reviewer #2**

*Discussion on manuscript title*. The authors are not consequent in their objectives: the model calculates the soil state (at present), but this is presented as soil change (a.o. in the title). For "change" you need (minimally) 2 values in time, these are the initial value and the final/current value. As the authors state there is currently no role of the parent material in their functions, there is no initial value and thus there can be no "change" calculated. Following the discussion inside the manuscript on reasons for poor model fits, parent materials may enter in future versions of the model and then change may come in. Advice: do not suggest "change" when it is not calculated, certainly not in the title.

**Re: reviewer J. Phillips**

*On the bivariate normal density function*. The authors reply that other bivariate distributions are possible, but this does not answer the concern (also by myself) that multivariate pdf could be considered as well. Advice: Add this to the announced change near lines 134-136 and comment on possible problems that this might introduce.

*On lines 144-154*. The authors state to disagree with the reviewer "as the present approach is effective at prediction soil property across wide variety of environments and ecosystems". Such effectivity was not present at all sites (e.g. the hot climates in Barbados or Taiwan) and anyway, statistical effectivity of predicting a depth-weighted parameter does not validate assumptions on the presence, absence or mutual compensation of progressive and regressive processes, related to equifinality or pathway dependency. Advice: mention in the text that the correct prediction of depth-weighted soil properties does not inform on progressive and regressive processes that affect the depth distribution of these properties.

**Re: review by editor**

*On the delineation of the model domain*: In statistical studies such as these it is a good practice to indicate the domain for which the model was parametrized, for instance to prevent extrapolation. That was behind my question. At the moment the model is fit to a wider data set, it is a new model

with a new domain. The current domain seems to be restricted to soils little affected by humans, with little aeolian deposition and for the temporal extents of the chronosequences. Advice: It should be no problem summarizing this.

*Response to question 2*: The quoted text states that "taking into account the differences in past and present climate would likely diminish disparities" (etc). This statement assumes a perfect model which does not exist and I advise to weaken it because it is not supported by evidence.

*Response to question 3*: The authors state that age definition is not a problem and suggest it is more easily done than measuring the clay content. This refers to the chronosequences used in the study, where age was determined. But what if the model is to be applied elsewhere? A model that is only valid at the locations of parametrization is not very useful, and I think that the authors do aim at a wider geographical application domain. Thus it is a legitimate question to ask if age determination at such unvisited locations is not more costly than to measure the target parameter directly. Advice: comment on this issue.

*Response to question 6*:  Fertilizer input transferred to EEMT may not change EEMT much, but quite some soil properties may respond strongly (which is a purpose of fertilization). Perhaps not the physical soil properties that were looked at; here we may have another delineation of the model domain. Advice: state in the text that human agricultural impacts were not addressed in this study, and that adaptation of the model to include these impacts may ask for adaptation of the model structure as well as a re-parametrization.

*Response to section 3.1, bias*: You did likely not introduce bias by the sampling amongst the chronosequences, it is the available data that are biased. Advice: mention in the text that this is an unavoidable limitation.

*Response to remarks on predicting potential landscape evolution*: The authors are quite defensive in their response. Certainly the present model cannot predict potential landscape evolution including human impact, because human impact is not included as a factor. When it will be included, that's another paper. Also, authors should realize that they do not predict OC nor its feedback with erodibility. It is a model for some soil physical parameters and that does not make it a landscape evolution model. Potential landscape evolution addresses the future. Please comment on how EEMT can be estimated for future scenarios. Advice: be realistic in what the current model can do and what it cannot (yet) do.